# Winter Precipitation Measurements in New England: Results from the Global Precipitation Measurement Ground Validation Campaign in Connecticut

Brian Filipiak,<sup>1,2</sup> David B. Wolff,<sup>3</sup> Aaron Spaulding,<sup>4</sup> Ali Tokay,<sup>5,6</sup> Charles N. Helms,<sup>6,7</sup> Adrian M. Loftus,<sup>6</sup> Alexey V. Chibisov,<sup>3</sup> Carl Schirtzinger,<sup>3</sup> Mick J. Boulanger,<sup>3,8</sup> Charanjit S. Pabla,<sup>3,8</sup> Larry Bliven,<sup>4</sup> EunYeol Kim,<sup>9</sup> Francesc Junyent,<sup>9</sup> V. Chandrasekar,<sup>9</sup> Hein Thant,<sup>9</sup> Branislav M. Notaros,<sup>9</sup> Gustavo Britto Hupsel de Azevedo,<sup>10</sup> and Diego Cerrai,<sup>1,2</sup>

<sup>1</sup>Department of Civil and Environmental Engineering, University of Connecticut, Storrs, CT

<sup>2</sup>Eversource Energy Center, University of Connecticut, Storrs, CT

<sup>3</sup>NASA Goddard Space Flight Center, Wallops Flight Facility, Wallops Island, VA

O <sup>4</sup>Department of Civil and Environmental Engineering, Princeton University, Princeton, NJ

<sup>5</sup>Goddard Earth Sciences Technology and Research, University of Maryland Baltimore County, Baltimore, MD

<sup>6</sup>NASA Goddard Space Flight Center, Greenbelt, MD

<sup>7</sup>Earth System Science Interdisciplinary Center, University of Maryland, College Park, College Park, MD

<sup>8</sup>Science Systems and Applications, Inc., Lanham, MD

<sup>9</sup>Department of Electrical and Computer Engineering, Colorado State University, Fort Collins, CO

<sup>10</sup>Department of Mechanical and Aero-Space Engineering, Oklahoma State University, Stillwater, OK

Correspondence to: Brian Filipiak (brian.filipiak@uconn.edu)

Abstract: Winter precipitation forecasts of phase and amount are challenging, especially in Northeast United States where mixed precipitation events from various synoptic systems frequently occur. Yet, there are not enough quality observations of winter precipitation, particularly microphysical properties from falling snow or mixed phase precipitation. During the winters of 2021-2022, 2022-2023, and 2023-2024, the NASA Global Precipitation Measurement (GPM) Ground Validation (GV) program conducted a field campaign at the University of Connecticut (UConn). The goal of this campaign was to observe various phases of winter precipitation and winter storm types to validate the GPM satellite precipitation products. Over the three winters at UConn, a total of 40 instruments were deployed across two observing sites that captured 117 precipitation events, including 19 phase transition events as indicated by the PARSIVEL<sup>2</sup>. These instruments included scanning and vertically pointing radars, along with suites of in-situ sensors. In addition, an unmanned aircraft system has been deployed in 2023-2024. Here, an overview of the different field deployments, instrumentation, and the datasets collected are presented. To showcase the observations, this article features a wide-ranging set of measurements collected from the instrument suite for the 28 February 2023 storm, during which six to eight inches of snow accumulated at the two different observing sites. Also included is a discussion on how these observations can be combined with other datasets to validate ground-based and remote sensing measurements and highlight important atmospheric processes that impact winter precipitation phase and amount.

### 1 Introduction

45

The Northeast United States is exposed to hazardous winter weather with heavy snowfall, high winds, and freezing rain each year. Owing to its terrain and conducive meteorological conditions, including strong thermal gradients between the land and ocean, the east coast of the United States is susceptible to major winter storms (Maglaras et al 1995). Forecasting when and where precipitation will occur remains one of the key challenges for forecasters. Not only do forecasts have to accurately predict the amount and intensity of precipitation in the correct location but also must identify what phase of precipitation will occur in each location. If there are multiple precipitation phases occurring at once, forecasters must correctly determine the precipitation amount for each precipitation phase (Maglaras et al 1995; Ralph et al. 2005, Novak et al 2023). Subtle shifts in the environmental conditions over an area can create significant forecast changes and different impacts than forecasted (Novak et al. 2023).

In addition to forecasting challenges, there are issues with the reliability of observations for specific winter weather phenomena. Since the types of observations vary so much, it can be difficult to obtain trustworthy and accurate measurements particularly for snow and ice (Rasmussen et al. 2012; Hurwitz et al. 2020). These range from manual observations, in situ sensors, and remote sensing instruments that can be as far away as outer space. Manual observations can vary significantly because some may be reported by trained National Weather Service (NWS) cooperative volunteer observers or other trained observers, such as CoCoRaHS (Cifelli et al. 2005), whereas others can be reported by any person with a mobile app (e.g., mPING, Elmore et al. 2014) on their phone. This creates a challenge for processing what exactly was measured and how accurate said measurement was. While many in situ sensors can accurately observe winter weather, they still have limitations (Martinaitis et al. 2015). This mainly is an issue at Automated Surface Observing System, ASOS (NOAA 1998) for precipitation accumulations caused by undercatch, a phenomenon that occurs with precipitation gauges where the orifice size of the measuring device prevents an accurate measurement due to the restrictive collection area (Rasmussen et al 2012; Martinaitis et al 2015). In addition, non-heated tipping buckets are not effective for frozen precipitation where accumulations will not be measured until the ice melts, potentially days or multiple storms later. Heated tipping buckets can fix some of these issues, but they still are not as accurate for frozen precipitation as weighing bucket precipitation gauges (White et al. 2004; Greeney et al. 2005; Tokay et al. 2010). Retrievals from remote sensing instruments are helpful especially where surface observations are not available. However, they have larger uncertainties and require refinement, particularly for frozen winter precipitation (Kulie et al. 2010). One significant challenge for frozen precipitation is incorporating the variable particle shape, mass, and density (Brandes et al. 2007; Kulie et al. 2010; You et al. 2017), which affects scattering and extinction properties retrieved by these instruments. In general, snow measurements can be impacted by their environment more than other precipitation phases; there is an urgent need to improve spatial and temporal frequency of snowfall observations (Doesken and Robinson 2009; Hurwitz et al. 2020). Many environmental variables, such as wetbulb temperature, particle size and fall speed, and snow density, contribute to identification of precipitation phase and

forecasting snowfall amounts (Rasmussen et al. 2012; Hurwitz et al. 2020), but they are not measured by any operational instruments.

70

A combination of three observational platforms is ideal to have a robust estimate of precipitation phase and amount with the given uncertainty of each platform. This is feasible with targeted field campaigns, which have contributed to improving forecasters' understanding of thermodynamics and microphysical processes (Novak et al. 2023). To this point, a suite of *in situ* and remote sensing instruments were deployed at the University of Connecticut (UConn) in Northeast United States as a part of a NASA Global Precipitation Measurement (GPM) Ground Validation (GV) field campaign. The timing of this GPM GV campaign allowed for synergistic overlap with the Investigation of Microphysics and Precipitation of Atlantic Coast-Threatening Snowstorms field campaign (IMPACTS, McMurdie et al. 2022) in the winters of 2021-2022 and 2022-2023. During the overlapping field deployment seasons, three snowfall events were sampled by instruments from both campaigns by having airborne observations of flight legs over or near the UConn site to be combined with remote sensing and ground-based observations at or around UConn. The length of the field campaign, over four months, allowed for the collection of a rich data set from numerous rain, snow, and mixed phase transition events.

The GPM mission (Hou et al. 2014, Skofronick-Jackson et al. 2017) was established to unify and advance precipitation measurements from space-borne radar and passive microwave sensors. This is an extremely important mission worldwide because GPM can measure precipitation in areas without observational networks and over the oceans. GPM presented an improvement over its predecessor, the Tropical Rainfall Measurement Mission (TRMM), because of its expansion to include the mid latitude region and ability to detect light precipitation rates and snowfall, albeit not without its challenges (Casella et al. 2017; Milani and Kidd 2023). The GPM mission has a core observatory along with a broader constellation of satellites. The core observatory is equipped with the Dual-Frequency Precipitation Radar (DPR) and the GPM Microwave Imager (GMI), a passive microwave radiometer across multiple frequencies. These spaceborne remote sensing instruments can observe a large portion of the globe (±68° of latitude) during both day and night which makes it especially useful for winter weather observations in the midlatitudes.

While there have been several GPM GV field campaigns across the United States and the globe, only a few have focused on winter weather. These include: LPVEx, the Light Precipitation Validation Experiment (Petersen et al. 2011) in Finland during the fall of 2010; GCPEx, the GPM Cold-season Precipitation Experiment (Skofronick-Jackson et al. 2015) in central Ontario, Canada during the winter of 2011-2012; OLYMPEX, the Olympic Mountains Experiment (Houze et al. 2017) in the Olympic Peninsula of Washington in the winter of 2015-2016; and ICE-POP, International Collaborative Experiment for PyeongChang Olympic and Paralympics (Petersen et al. 2018) in South Korea during February 2018. There have been a few recent field campaigns (not GPM GV related) that have been undertaken in Northeast United States, (OWLES – Kristovich et al. 2017, IMPACTS, and WINTRE-MIX – Minder et al. 2023), but these focused on specific types of winter phenomena: lake-effect snow, snowstorms, and precipitation phase, respectively.

Between 1980 and 2023, there have been 22 winter storms that caused at least \$1 billion of damage in the Continental United States (Consumer Price Index-adjusted); the total estimated loss for the 22 events was \$97.2 billion

(NOAA NCEI 2023). It is important to note that 21 of these 22 events were in the Northeast Climate Region, which has a high concentration and frequency of winter storms compared to the rest of the United States (Changnon 2007; Lazo et al. 2020; NOAA NCEI 2023). With the UConn site being located in southern central part of the Northeast United States, it is in an ideal location, due to being far enough inland that colder air is more prolonged when winter storms happen, to sample a wide variety and intensity of winter precipitation that previous winter GPM GV campaigns did not experience.

This article presents an overview of the GPM GV campaign at UConn for which the main objectives were to collect a rich and consistent set of observations for winter precipitation to validate and improve GPM based retrieval algorithms during winter storms in the Northeast United States. With 117 precipitation events across the three winters, a myriad of winter weather phenomena was captured ranging from heavy snow to freezing rain to ice pellets to events with multiple phase transitions. This unique dataset can be used for examining the GPM's retrieval algorithms for winter weather, validating precipitation phase algorithms or numerical model simulations, and analysing physical processes that can explain key components of winter phenomena, such as snow microphysics. Section 2 presents an overview of the different field deployments and instrumentations. Section 3 details the data processing for each instrument, which includes calibration or validation notes. Section 4 highlights an impactful snowstorm as case study from the 2022-2023 winter to spotlight the variety of observations from the instrumentation. Section 5 details the data availability; Section 6 closes with concluding remarks and points to directions of future research.

# 2 Deployment Sites and Instrumentation

A suite of ground-based instruments was assembled with the objective of collecting observations of winter precipitation, improving satellite retrieval algorithms, and understanding the many physical processes and properties of winter weather phenomena. These observations aimed at providing a more complete picture of microphysical and thermodynamic processes that occur near the surface or in the lowest levels of the troposphere. Since it takes significant resources to have such a complete set of instrumentation, including many repetitive measurements across instruments, it was important to obtain as many observations as possible. The location of the UConn sites aligns generally within the swath of many of the winter storms that occur in the Northeast United States (Fig. 1a) as shown previous climatological studies (Miller 1946, Davis et al. 1993). UConn's inland location provides the opportunity for prolonged periods of below freezing temperatures for storms that pass to the south and east. Additionally, storms that move from southwest to northeast by UConn allow for the opportunity for phase transitions due to the location of the warm and cold fronts.

The instruments were deployed over three separate winter seasons, 2021-2022, 2022-2023, 2023-2024. Two separate locations were used during the three years of deployments and will be labeled as GAIL and D3R. The GAIL site (41°48'28"N, 72°17'38"W, elevation of 149 m) and the D3R site (41°49'04"N, 72°15'27"W, elevation of 213 m) are close in distance, approximately 3.2 km apart, but the local terrain could significantly impact the observed precipitation. The GAIL site was located near the base of the Willimantic River valley, whereas the D3R site was located at the top of a hill

above the river valley; the 64 m elevation difference could cause differences in precipitation onset, phase, and amount to vary across the two sites (Fig. 1b). Because the goal of the observations collected by the instrument suites aims to improve the GPM satellite retrievals, all the instruments provide measurements that can be directly compared to the GPM satellite retrieval algorithms output or be used in the development or refinement of the algorithms. In the subsequent subsections, a description of each instrumentation and location by deployment year.

Fig. 1. a) A map of Northeast United States showing location of UConn relative to select cities is on the left; on the right, the map shows the location of the UConn sites. The circle outlines the scanning radius of the D3R radar, 38 km, and the green dashed line shows the direction of the Range Height Indicator (RHI) scans from the D3R b) West to East elevation cross section between the GAIL and D3R observing sites at UConn used during GPM GV field campaign.

### 145 **2.1 Instrumentation**

# 2.1.1 All-in-One 2 (AIO)

The All-in-One 2 (AIO, Fig. 2) is a sonic weather sensor, produced by Met One, that measures wind speed and direction, ambient air temperature, relative humidity, and barometric pressure, thereby providing the necessary background context for precipitation observations. The temperature and relative humidity sensors are unaspirated and filtered. AIO collects high resolution data at 1 Hz; it has an accuracy of +/- 0.2 °C for ambient air temperature and of +/- 3 % for relative humidity, which is essential for understanding precipitation phase. Observations from the AIO are recorded as 1-minute averages. The observations from the AIO can enhance the interpretations and limitations of precipitation observations, such as snow microphysics (King and coauthors 2024). The GPM goal is to observe global precipitation; thus, the ability to constrain the observed meteorological conditions can enhance and validate the end outcome (how much of each precipitation phase fell) and the important quantities remotely sensed by the satellite (reflectivity, precipitation rate, particle size, etc.).

Fig. 2. Deployment of the Met One All-in-one 2 during 2022-2023 at the D3R site.

### 2.1.2 Micro Rain Radar (MRR)

The Micro Rain Radar (MRR, Fig. 3) is a 24 GHz K-band vertically pointed profiling radar, with a heated radar dish, produced by METEK. It is effective at sampling the lowest levels of the atmosphere and identifying doppler spectra of hydrometeors, which can be used to identify bright brands that indicate falling snow particles melting to rain droplets. Two different METEK MRRs were used during the field deployments.

In 2021-2022, an MRR-2 was deployed. The MRR-2 has 32 range gates with a minimum detectable reflectivity of 2 dBZ at 1000 m (METEK MRR-2). During its deployment, the vertical resolution was set to 35 m and had a 30-second raw sampling frequency which was averaged to 1-minute, which allows for observations of the lowest 2 km of the atmosphere (the two range gates nearest the surface bins are not typically used due to potential ground clutter issues). Raw MRR-2 observations collected were for reflectivity, doppler velocity, and spectrum width with additional processing creating additional quantities such as rain rate and liquid water content. The MRR-2 was designed for liquid precipitation, but a commonly used post-processing algorithm (Maahn and Kollias 2012), can be applied to improve noise removal and add a de-aliasing component to provide more effective Doppler velocity, reflectivity, and spectral width specifically for frozen and mixed phase precipitation.

During 2022-2023 and 2023-2024, an MRR-Pro was deployed instead of the MRR-2; this was done due to constraints on the number of available instruments. The MRR-Pro is similar to the MRR-2 and had the same vertical range resolution but has additional range gates, set to 128 for both deployments, and a minimum detectable reflectivity of -8 dBZ at 1000m (METEK MRR-Pro). Originally in 2022-2023, the range gates were set to 256, but there were issues with the vertical velocity retrievals and was shifted to 128 to avoid this issue. The same sampling frequency and averaging periods were kept between campaigns to create a similar dataset. Unlike for the MRR-2, there is no additional software or algorithms to process MRR-Pro observations for frozen precipitation; METEK has added new internal processing steps to the MRR-Pro that has provided some improvements, but there remains a need for the creation of a processing software similar to Maahn and Kollias 2012 for the MRR-Pro.

Fig. 3. Deployment of the MRR-Pro during 2023-2024 at the GAIL site.

# 2.1.3 Pluvio<sup>2</sup> Weighing Gauge (Pluvio)

The Pluvio<sup>2</sup> Weighing Gauge (Pluvio, Fig. 4) is a weighing precipitation gauge manufactured by OTT Hydromet designed to collect all phases of precipitation. Pluvio records precipitation amount and intensity in both real time (output delay of 20 seconds), and non-real time (output delay of 5 minutes). With a sampling interval of 1 minute, the accuracy of precipitation amount is +/- 0.05 mm over a 60-minute measuring window and +/- 0.1 mm min<sup>-1</sup> for precipitation intensity. Two types of Pluvio were deployed during the campaign with one being a 200 cm<sup>2</sup> collection area and the other being a 400 cm<sup>2</sup> collection area. The 200 cm<sup>2</sup> model was heated during its deployment. Pluvio's can be subjected to the phenomenon of undercatch due to the orifice limiting accurate totals (Rasmussen et al 2012; Martinaitis et al 2015), but a double alter wind shield was deployed around it to reduce this phenomenon. In addition, the redundant measurements with other instruments can be used to validate the observations. The measurements from the Pluvio can be used to calibrate the GPM satellite or ground-based radar retrieval algorithms for precipitation amount and rate.

Fig. 4. Deployment of the Ott Pluvio<sup>2</sup> Weighing Gauge during 2022-2023 at the D3R site.

#### 2.1.4 Platform for In situ Estimation of Rainfall Systems (PIERS+)

# 2.1.4.1 PARSIVEL<sup>2</sup>

PARSIVEL<sup>2</sup> (Particle Size Velocity), a laser-based disdrometer produced by OTT Hydromet, was primarily developed to measure the size and fall velocity of raindrops (Loeffler-Mang and Joss 2000) but can also measure the size and fall velocity of snowflakes (Loeffler-Mang and Blahak 2001) with limited accuracy. The PARSIVEL<sup>2</sup> is at the center of a NASA designed Platform for In situ Estimation of Rainfall Systems (PIERS+) platform shown in Fig. 5. Particles passing through the laser beam (30 mm wide, 1 mm high, 180 mm long) of the PARSIVEL<sup>2</sup> cause voltage decrease by light extinction. The amplitude and duration of induced voltage signal is related to particle size and fall velocity, respectively. PARSIVEL<sup>2</sup> raw output is 32 by 32 size and fall velocity matrix. Due to sensitivity of laser device to the small particles with minimum detectable size 0.25 mm in equivalent diameter, the first two size bins in the PARSIVEL<sup>2</sup> observations are always empty. In PARSIVEL<sup>2</sup> the bin width increases from 0.125 mm to 3 mm with size due to the increasing uncertainty of particle size measurement. Therefore, it is not reliable to measure the maximum drop diameter at high accuracy in moderate to heavy rain (> 5 mm h<sup>-1</sup>), and the complex shape of falling snowflakes can be an issue to measure the size accurately. PARSIVEL<sup>2</sup> is also be used to determine present weather and visibility based on fall velocity versus size measurements, which can give synoptic weather codes similar to ASOS stations. Processed PARSIVEL<sup>2</sup> observations can give important microphysical properties like particle size distributions (PSD), liquid water content, and number concentration.

Fig. 5. Deployment of the Platform for In situ Estimation of Rainfall Systems (PIERS+) during 2023-2024 at the GAIL Site. PIERS+ includes the OTT PARSIVEL<sup>2</sup>, two Met One tipping buckets, and a RM Young Anemometer.

### 2.1.4.2 Tipping Bucket Rain Gauges

Tipping Buckets are a pair of Met One 380 rain gauges deployed on each PIERS+ platform on either side of PARSIVEL<sup>2</sup>. The gauges are made of metal and are used to measure rain only from GPM GV activities as these gauges are not heated. Each gauge utilizes a dual-chambered tipping bucket that is self-emptying via a hole in the bottom of the mechanism. Each tip of the bucket indicates a rain accumulation of 0.01 inch or 0.25 mm. As the bucket tips, it logs the time to the nearest second which can be recorded on a logger or other device. Similar to the Pluvio, the tipping buckets record precipitation amount and intensity and can be subjected to undercatch. However, the pair of buckets allow for verification of the others' observations, which is helpful for identifying instrument errors.

### 2.1.4.3 RM Young Mechanical Anemometer

Attached to the PIERS+ platform is a mechanical RM Young anemometer which is used to measure wind speed and direction. This instrument provides redundancy to the AIO sonic anemometer and allows for verification between the two sensors. The mechanical anemometer has an accuracy of +/- 0.3 m s<sup>-1</sup>, with a minimum speed of 1.0 m s<sup>-1</sup>, and +/- 3° for direction. The sampling frequency was 1 Hz. Since only one PIERS+ platform was deployed in 2022-2023 and 2023-2024, there was only one mechanical anemometer each deployment year.

#### 2.1.5 Precipitation Imaging Package (PIP)

The Precipitation Imaging Package (PIP, Fig. 6, Pettersen et al. 2020, 2021; Helms et al. 2022; Tokay et al. 2023; King and Coauthors 2024) is a NASA-developed video disdrometer consisting of a high-speed camera pointed at a halogen light source. The light source, located 2 m away from the camera lens, acts to backlight the precipitation particles that fall through the sampling volume, which is located 1.33 m from the lens. Critically, PIP uses an open sampling volume that is sufficiently far from the instrument components to prevent contamination of the measurements by particle shattering and other effects of the flow around the instrument under typical wind conditions (Newman et al. 2009). The PIP images are 640 by 480 pixels with a calibrated pixel size of 0.1 mm by 0.1 mm and are continuously collected at a rate of 380 frames per second. Due to bandwidth limitations, the images are compressed in the vertical by averaging adjacent pixels resulting in an effective pixel size of 0.1 mm by 0.2 mm. Although the actual sampling volume dimensions are a function of particle size (Newman et al. 2009), the sampling volume is no greater than 64 mm by 48 mm in the viewing plane of the camera and has a depth of field of approximately 117 times the maximum particle dimension (0.23 m for a 2 mm). The PIP analysis software generates a wide variety of direct particle measurements, although the two key measurements are the equivalent diameter and vertical motion as these are used to produce the PSD and the particle density estimates. PIP produces multiple levels of data output, as described in King and Coauthors 2024, and its highest order processed observations include particle count, rain and non-rain rate, particle density, and particle fall speed.

Fig. 6. Deployment of the Precipitation Imaging Package during 2023-2024 at the GAIL site.

#### 2.1.6 Aerosol, Cloud, Humidity Interactions Exploring and Validation Enterprise (ACHIEVE)

The NASA-Goddard Space Flight Center Aerosol, Cloud, Humidity Interactions Exploring and Validation 250 Enterprise (ACHIEVE, Fig. 7) is a transportable ground-based laboratory for obtaining remotely sensed measurements of cloud and precipitation properties (Loftus et al., 2016). ACHIEVE's primary instrument is the scanning W-band (94 GHz) Doppler cloud radar, manufactured by ProSensing Inc., and provides high-resolution measurements of reflectivity, Doppler velocities, Linear Depolarization Ratio (LDR), Doppler spectrum width, as well as Doppler power spectra. ACHIEVE was deployed in the 2022-2023 campaign at the D3R site. During the deployment, the W-band radar operated mostly in vertically 255 pointing mode in order to capture the evolution of cloud and precipitation system vertical structures as they passed overhead. During active precipitation periods, additional Range Height Indicator (RHI) scans were performed every 10 minutes over the GAIL site to the west as well as N-S oriented in an effort to sample spatial variability of the systems and for obtaining near-coincident data with the D3R scanning radars. Coincident RHI scans along the ER-2 flight paths were also conducted during overflights on 28 February 2023. The W-band radar collected data from 11 precipitation events, six of which were 260 classified as phase transition events (rain to snow or snow to rain), and one mixed phase event with graupel and freezing rain.

Fig. 7. Deployment of the ACHIEVE trailer with the W-Band Radar on the roof during 2022-2023 at the D3R site.

# 2.1.7 Dual-frequency Dual-polarized Doppler Radar (D3R)

The ground-based Dual-frequency, Dual-polarization Doppler Radar (D3R, Fig. 8; Chandrasekar et al. 2012; Vega et al. 2014) was specifically designed to match the Ka and Ku frequencies of the GPM DPR with its Ka band operating at 35.56 GHz +/- 25 MHz and the Ku band at 13.91 GHz +/- 25 MHz. During its deployment in the winter of 2022-2023, the radar operated with a range of 38 km (shown in Fig. 1a) with 150 m range resolution. With this configuration, the minimum detectable signal at 15 km is -8 dBZ for the Ku band and -3 dBZ for the Ka band. The D3R operates like a more traditional radar with both plan position indicator (PPI) and RHI scans compared to the MRRs and ACHIEVE; the PPI scans covered the areas WSW to ENE of the radar location, which includes the area over the GAIL site. Individual RHIs in the direction of the GAIL site, as well as true north, were also conducted to match the ACHIEVE. Besides its observations of reflectivity and doppler velocity, the dual-polarization nature of the D3R allows for the hydrometeors to be observed with greater detail with

a suite of dual-polarization products. The D3R's ability to provide dual-polarized derived products is necessary for understanding the microphysical characteristics of the falling particles.

Fig. 8. Deployment of the Dual-frequency, Dual-polarization Doppler Radar during 2022-2023 at the D3R site.

#### 2.1.8 Snowflake Measurement and Analysis System (SMAS)

The Snowflake Measurement and Analysis System (SMAS, Fig. 9) is a novel system for measurement and analysis of snow particles in freefall, designed, developed, and built in the Electromagnetics Laboratory at Colorado State University (CSU). The main features of the SMAS are: seven high-resolution cameras strategically placed in a 3D fashion for reconstruction of shapes of snowflakes based on photographs of particles in freefall from multiple views (Kleinkort et al. 2017), capability for measurement and analysis of multiple snowflakes at once, namely, processing of images with multiple snowflakes per frame, and comprehensive characterization of microphysical properties of snowflakes. Some parameters of the SMAS are 1440-cm³ imaging volume, 5-Megapixel cameras, and 31-µm pixel resolution (Thant et al. 2022). The instrument is also equipped with six flashes, five sensor boards, two lasers, a printed circuit board, which triggers the

cameras and flashes when a snowflake falls through the system and the laser plane gets blocked, and a built-in computer, which organizes and stores the images captured by the cameras.

Measured and estimated particle properties are fall speed, density, effective dielectric constant (Notaroš 2021), and scattering observables (Chobanyan et al. 2015). The SMAS is also capable of calculating the PSD (Huang et al. 2017), e.g., the particle cross section area and diameter, respectively, in real time. Multiple studies have shown that neural networks, can be applied to classify snowflakes into six geometrical categories (small particle, planar crystal, columnar crystal, combination of columnar and planar crystal, aggregate, and graupel), as well as five separate degrees of riming can be observed, which is essential for understanding the formation and meteorological environment of snow (Hicks and Notaroš 2019, Key et al. 2021, Thant et al. 2023a).

Fig. 9. Deployment of the Snowflake Measurement and Analysis System during 2023-2024 at the GAIL site.

# 2.1.9 CSU-Modified Multi-Angle Snowflake Camera (MASC)

The Multi-Angle Snowflake Camera (MASC, Fig. 10; Garrett et al. 2012), a commercially available instrument, uses three cameras in the horizontal plane separated by 36° to capture high-resolution photographs of snowflakes or other frozen hydrometeors in freefall from three views, while simultaneously measuring their fall speed. The pixel resolution is 50 μm, the virtual measurement area is 30 cm², and the measurement volume rate is ~200 cm³ s⁻¹. The instrument includes two near-IR emitter-receiver pair arrays positioned one above the other, and as a particle falls through the lower array, it triggers all cameras. Additionally, the particle fall speed is obtained from the fall time between the two triggers. To enable 3D shape reconstruction, we added two cameras to the CSU-MASC, "externally", above the original cameras and at a 55° angle with

respect to the horizon, to provide additional views (Kleinkort et al. 2017, Notaroš 2021). Additionally, we can estimate PSD (Huang et al. 2017) and classify snowflake geometry, riming degree, and melt/dry state (Hicks and Notaroš 2019, Key et al. 2021, Thant et al. 2023b).

Fig. 10. Deployment of the Multi-Angle Snowflake Camera during 2023-2024 at the GAIL site.

### 2.1.10 WxUAS

The Oklahoma StormTrooper is an Unmanned Aerial Weather Measurement System (WxUAS, Fig. 11) capable of sampling the low-altitude wintry precipitation environment, including mix-phased and freezing precipitation (Britto Hupsel de Azevedo 2024; Azevedo et al 2024; Azevedo et al. 2025). This WxUAS provides in-situ samples of pressure, temperature, relative humidity, and particle size counts. Additionally, it provides remote samples of reflectivity and velocity from an onboard, vertically pointing 74GHz millimeter wave (mmWave) radar with 0.4887m resolution (see Table C.1 of Britto Hupsel de Azevedo 2024 for radar specifications). Leveraging the mobility of UAVs, the proposed WxUAS creates a spatial distribution of the measured atmospheric parameters. Via repetition of the flight pattern at regular intervals, the StormTrooper captures the temporal evolution of the measured parameters' spatial distribution. Together, the spatiotemporal aspects of this WxUAS' measurements produce insight into atmospheric mechanisms that govern the low-altitude wintry precipitation environment.

Temperature, relative humidity measurements and particle counts are produced in an actively aspirated sensor shield at 1 Hz. Due to its shape, this shield design can leverage gravity to mechanically separate large precipitation drops from the air. This separation occurs because larger droplets are dominated by their inertia while small droplets will follow the streamlines on the intake system according to Stokes flow (Britto Hupsel de Azevedo 2024; Azevedo et al 2024; Azevedo et al. 2025). This flow characteristic allows the system to sample the atmospheric conditions without exposure to harmful freezing precipitation. The 74 GHz, vertically pointing, phased array radar can use beam forming techniques to compensate aircraft motion, making Doppler velocity an approximation of hydrometeor fall velocity. WxUAS provides a unique perspective that connects the ground instrumentation with remote sensing instruments collecting observations above ground level. Its utility in capturing thermodynamic changes in the lowest levels of the atmosphere is essential for understanding precipitation events in near freezing conditions.

Fig. 11. The Oklahoma StormTrooper: an Unmanned Aerial Weather Measurement System (WxUAS) with its remote (left) and in situ (right) payloads. Deployed during 2023-2024 at the GAIL site.

#### 2.1.11 Ceilometer

Ceilometers are useful tools for observing overhead cloud and visibility conditions and can provide additional information about the atmospheric boundary layer. This ceilometer, produced by Vaisala (Fig. 12), uses pulse diode laser Light Detection And Ranging, LIDAR, technology at 910 nm wavelengths; the vertical observed profiles are able to measure

aerosol backscatter up to 15 km, and retrieve cloud base heights and boundary layer structure up to 4.5 km. These measurements can be compared to estimates of cloud base levels from the WxUAS, which provides validation and context of the vertical profiles it has collected. Observations of cloud layers can also be compared to ground-based radar observations for additional insight.

Fig. 12. Deployment of the Vaisala Ceilometer during 2023-2024 at the GAIL Site.

### 2.2 Deployment Sites

### 2.2.1 2021-2022

During the first winter of the campaign, the GAIL site was used to host all the instrumentation. A core set of instrumentation was identified to be located at UConn during this and all future winters. This set consisted of: (i) an MRR (MRR-2 was used in 2021-2022 only), (ii) a Pluvio, (iii) an AIO, (iv) a PARSIVEL<sup>2</sup>, and (v) a PIP. This core

instrumentation provided comparable and consistent measurements year to year and collected a wide array of observations from microphysical observations, precipitation amounts, meteorological conditions, and ground-based radar. The instruments were deployed from December 1, 2021, to April 26, 2022.

#### 2.2.2 2022-2023

For the second deployment, the goal was to increase the diversity of instrumentation as well as create an additional deployment site at a higher elevation. This led to instruments being located at both the GAIL and D3R sites with the same core instruments at both sites, except for the MRR-2 being replaced by an MRR-Pro. PIERS+ were also added to the core instrumentation set since there is a lack of quality-controlled rain gauge observations near UConn. The additional precipitation measurements were also valuable during the warm months of the deployment because they could be compared with the Pluvio for further validation. At the D3R site, the D3R and the 94 GHz W-band radar mounted on the ACHIEVE trailer were co-located to allow for comparable radar observations. At the GAIL site, additional instruments to measure microphysical properties of snow were included. The SMAS and MASC collected observations to help verify and enhance the observations from the PIP. The instruments were deployed at UConn from December 23, 2022, to April 11, 2023.

#### 2.2.3 2023-2024

During the 2023-2024 deployment, the goal was to further validate the instrumentation used in previous deployments, as well as to expand the vertical atmospheric profile measurements. To do this, only the GAIL site was used, and two sets of each core instrumentation was deployed in close proximity to one another; this technique has been used before in other field campaigns and classified as a "supersite" (Gultepe et al. 2019; Kim et al. 2021). In addition to the extra instrumentation, a Vaisala CL-51 ceilometer was deployed to provide measurements essential for comparison with the low-altitude WxUAS. The WxUAS is a novel tool to sample the lowest levels of the atmosphere at quicker intervals than would be possible with radiosondes. The WxUAS was deployed in the style of Intense Observing Periods (IOPs) as opposed to the instruments, which were collecting observations continuously. The instrumentation was deployed from December 15, 2023, to May 21, 2024. For an overall summary, Table 1 summarizes the instrumentation deployed during each campaign; Fig. 13 showed photographs of how the deployment sites were setup in 2022-2023 and 2023-2024.

| Instrument (owner)           | 2021-22  | 2022-23    | 2023-24 |
|------------------------------|----------|------------|---------|
| MRR-2 (NASA)                 | ✓        | Х          | Х       |
| Pluvio (NASA)                | <b>√</b> | <b>√</b> * | √ (x2)  |
| All-in-One-2 (NASA)          | <b>√</b> | <b>√</b> * | √ (x2)  |
| PARSIVEL <sup>2</sup> (NASA) | <b>√</b> | <b>√</b> * | √ (x2)  |

| PIP (NASA)             | ✓ | <b>√</b> * | √ (x2) |
|------------------------|---|------------|--------|
| D3R (NASA)             | Χ | √D         | Х      |
| ACHIEVE (NASA)         | X | √D         | Х      |
| MRR-Pro (NASA)         | Х | <b>√</b> * | √ (x2) |
| Tipping Buckets (NASA) | Х | <b>√</b> * | √ (x2) |
| SMAS (CSU)             | X | √G         | ✓      |
| MASC (CSU)             | Х | √G         | ✓      |
| RM Young (NASA)        | X | √D         | ✓      |
| Ceilometer (NASA)      | Х | Х          | ✓      |
| WxUAS (OSU)            | Х | Х          | ✓      |

Table 1. List of instruments deployed during the three winters of the NASA GPM GV field campaign at UConn. The \* in the 2022-2023 column indicates the instrument was deployed both at the GAIL and D3R sites; the superscript G in the 2022-2023 column indicates the instrument was only at the GAIL site; the superscript D in the 2022-2023 column indicates the instrument was only at the D3R site. The (x2) in the 2023-2024 column indicates there were two instruments deployed at the GAIL site.

Fig. 13. a) Deployment of instruments at D3R site in winter of 2022-2023; b) Deployment of instruments at GAIL site in winter of 2023-2024; c) Deployment of instruments at GAIL site in winter of 2023-2024 with WxUAS in the foreground.

### 3 Data Processing and Quality Control

In this section, a list of precipitation events across the three years of deployments are derived from the PARSIVEL<sup>2</sup> to provide a starting point for specific research applications (Appendix A). To accompany the event list, additional in-depth descriptions are given for the quality control or data collection on certain instrumentation. All datasets come with a base level documentation about the quality control performed, but here, more in depth analysis and explanations are provided for non-standard quality control or data collection practices.

#### 390 3.1 Precipitation Events

Over the three-year deployment, 117 unique precipitation events were observed at the UCONN site which includes 19 snow only events, 79 rain only events, and 19 mixed precipitation phase events. Event start and end times were identified based on the PARSIVEL<sup>2</sup> observations; the event phase is determined based on the present weather algorithm. Events are defined as periods of precipitation separated by greater than 60 minutes of no precipitation. Table 2 lists out the number of

events by phase and year; Appendix A contains the full list of events and times. Mixed phase events can have one or more phase transitions between rain and snow or other frozen precipitation phases.

| Year and Site     | Rain Events | Snow Event | Mixed Phase Events |
|-------------------|-------------|------------|--------------------|
| 2021-2022 at GAIL | 36          | 8          | 6                  |
| 2022-2023 at GAIL | 15          | 6          | 6                  |
| 2022-2023 at D3R  | 15          | 5          | 6                  |
| 2023-2024 at GAIL | 28          | 5          | 7                  |

Table 2. Summary of the number of events by precipitation phase and year based on the full list from Appendix A.

#### 3.2 Data Quality Control and Processing

# 400 3.2.1 All-in-One 2 (AIO)

The AIO provides a standard collection of surface meteorological observations to help provide context for the precipitation and microphysical measurements from other instruments. Of the five key measurements it records (temperature, relative humidity, surface pressure, wind speed and direction), temperature, surface pressure and wind speed required additional quality control based on initial comparisons to other nearby surface stations. Relative humidity was too difficult to quality control because of the distance to the nearest reference surface observations; wind direction quality control could be connected to the wind speed since invalid wind speed observations likely would have invalid direction measurements. In addition to the quality control of the observations, each year of data collected was also checked from missing observation minutes; these were filled in during the quality control with -9999 and a summary table of the AIO deployments is available in Table 3.

| AIO Year and Site        | Start Time        | End Time         | Percent of Missing Observations |
|--------------------------|-------------------|------------------|---------------------------------|
| 2021-2022 GAIL Roof      | 12/7/2021 15:20Z  | 4/26/2022 18:30Z | 0.03 %                          |
| 2022-2023 GAIL Roof      | 12/23/2022 21:51Z | 4/11/2023 17:43Z | 1.08 %                          |
| 2022-2023 at D3R Ground  | 12/23/2022 20:48Z | 2/27/2023 17:40Z | 0.06 %                          |
| 2023-2024 at GAIL Ground | 12/16/2023 14:28Z | 5/17/2024 12:40Z | 0.02 %                          |
| 2023-2024 at GAIL Roof   | 12/16/2023 14:33Z | 5/21/2024 13:40Z | 0.05 %                          |

Table 3. Summary of the AIO deployment including start and end times along with the percentage of missing observation minutes.

To verify the AIO observations, the reference data used for comparisons were from the 1-minute ASOS observations at Willimantic, CT (KIJD), daily climate records from the Storrs, CT Global Historical Climate Network (GHCN) site, and the RM YOUNG anemometer at the D3R or GAIL sites. Temperature and pressure were validated using

the ASOS observations as that is the nearest surface observations to the deployment location; due to the chaotic nature of wind measurements, the RM Young anemometer was used to quality control the wind speed observations due to it being colocated with the AIO. The KIJD ASOS site is located to the SE of the deployment sites at 76 m elevation and 10.5 km away from D3R and 11.7 km away from GAIL. The Storrs GHCN site is located to the SE of the deployment sites at 202 m elevation and 3.5 km from D3R and 5.6 km away from GAIL.

For the quality control of the surface pressure, all observations were converted to sea level pressure to remove the bias due to the elevation differences. Outlier sea level pressure values, outside the range of 920 to 1050 hPa, were removed and marked as erroneous; the remaining values were used to calculate the bias and standard deviation for each respective year. Additional outliers were identified by identifying periods where the differences between the AIO and KIJD outside the bias +/- 2 standard deviations. All outliers were manually checked against the KIJD observations. If the outlier occurred for only a 1-minute period, then the value was corrected using linear interpolation of the observations 5 minutes on either side of the outlier. All outliers that were erroneous values were changed to -9999.

For the wind speed, initial plots were made of the distribution of wind speeds from the AIO and RM Young; there was a clear high bias in the AIO and produced numerous outlier observations. All wind speeds more than 40 m s<sup>-1</sup> were marked as erroneous. Based on these distributions, two thresholds were set based on whether the AIO was located on the ground, or on the roof of a trailer with the threshold being 20 m s<sup>-1</sup> for the ground and 25 m s<sup>-1</sup> for the roof locations. These were determined because both the RM Young and ASOS did not report 1-minute wind speeds in excess of 20 m s<sup>-1</sup> over the three-year period, and an extra 5 m s<sup>-1</sup> was added for the roof locations due to the increase in wind speed with height above the ground level. All suspect and erroneous values were checked manually for consistency as it is possible to get higher wind gusts. Upon completion of this process, suspect values that were determined to be erroneous were changed and marked as so. All erroneous values were set to -9999, and the suspect values were flagged with the original values retained.

For the quality control of the temperature observations, most of it was done manually due to the changes of the different elevations of the sites, the local terrain, and the diurnal nature of the temperature cycle. With the local topography of the Willimantic Valley, it was possible to have a wide range of temperatures particularly at night with the ASOS site at the bottom of the valley, GAIL site on the valley slope, and the D3R site at the top of the valley. To provide some baseline for outlier observations, the mean bias and standard deviation were calculated between the AIO and ASOS, as was done for pressure. The bias plus two standard deviations was then used to determine two thresholds based on the site location; the GAIL site had a threshold set of a difference of 3 °C, and the D3R had a threshold set for 4 °C. To aid the manual quality control, weekly timeseries were produced using the AIO and ASOS observations. Fig. 14a shows an example timeseries with the actual observations over a two-week period. A second plot was made (not shown) of the difference between the two observations. Any periods with differences greater than the threshold were marked down to be investigated further. The GHCN site in Storrs, CT, was used to help determine did the diurnal cycle seem to match in terms of maximum and minimum temperatures. This was especially helpful if the sites had significant differences between them as the GHCN site is more closely located and at a closer elevation to the deployment sites.

One issue to note with the temperature observations was in periods after the computer connected to the AIO was restarted, the temperature would not produce negative temperature observations right away. Fig. 14a shows an example of this during 2023-2024, where the KIJD records temperatures well below freezing but the AIO is reporting temperatures of 5-10 °C above freezing. The GHCN data was useful in verifying if the minimum temperature went below freezing or not on a given day. Periods when this occurred were manually identified and corrected by multiplying the AIO observations by a value of -1. Fig. 14b shows the absolute values of both temperature time series over the same time window, and the magnitude both of observations match further confirming sets the issue with

Fig. 14. Example of timeseries from December 2023 used in the quality control of the AIO; a) timeseries of temperature measurements from the AIO and KIJD ASOS; b) the absolute value of the temperature observation from AIO and KIJD ASOS.

#### 3.2.2 Micro Rain Radar (MRR)

Two varieties of MRRs, as described in Section 2.1.2, were used during the three deployment winters. The MRR-2 has a commonly used post-processing algorithm (Maahn and Kollias 2012), which enhances the radar returns in frozen and mixed precipitation. This can be done to improve noise removal and add a de-aliasing component to provide more effective doppler velocity, reflectivity, and spectral width. This algorithm could be applied to the MRR-2 data in 2021-2022. However, the MRR-Pro, which was used in the following two deployments, does not have a generally accepted method to improve the radar returns in frozen or mixed precipitation akin to Maahn and Kollias 2012. New approaches to provide improved radar data are being developed, and here, the MRR-Pro data was processed through a new method detailed in Williams 2023. This approach has two versions: a calibrated version with velocity de-aliasing and recalculated moments and

a disdrometer-calibrated version using a surface disdrometer, which follows a similar approach. The general approach removes range dependent noise and interference to find the lowest clutter free range gate and then applies a clutter filter adjustment to the clutter free range gates; from there, the spectra is reprocessed so velocity de-aliasing can be applied. Here is where the two methods differ as the calibration can be done compared to the originally observed spectra or against a surface disdrometer. In Section 4, we will present MRR-Pro data using this method with the calibration against the original spectra.

# **3.2.3 WxUAS**

As detailed in Section 2.2.3, the StormTrooper WxUAS was deployed in two IOPs in March 2024. In these deployments, the WxUAS sampled two storms targeting mixed-phased precipitation often seen in the temperature and pressure gradients caused by the leading and trailing edges of these storms. To increase the chances of capturing the lowaltitude phase transition periods while accommodating the limited flight crew size and the FAA crew resource management practices, the WxUAS deployment plan was based on an 8-hour window. The intention was to start sampling one to two hours before the target storm effects were forecasted to occur and end it another one to two hours after. For the approaching storm cases, the targeted deployment would characterize the local conditions before the arrival of the storm's frontal boundaries, its arrival, and its transition across boundaries. For departing storms, this same strategy would characterize the local conditions during the storm, its departure, and the conditions behind its trailing edge. Considering the timescale in which precipitation phase changes had been reported by the surface observations in the past three winters and the time necessary to perform the pre- and post-flight safety procedures, the 8-hour WxUAS data collection window was broken down into flights every 15 minutes. For these profiles, the WxUAS maintained a 3 m s<sup>-1</sup> ascent rate while countering the winds to hold latitude and longitude over its take-off spot near the surface instruments. The targeted altitude for the WxUAS profiles was 900 m (approximately 3000 ft.). However, given the FAA's 120 m flight ceiling for commercial UAV flights, an altitude restriction waiver request was submitted. Even though the study site is in uncontrolled airspace with low traffic, particularly when hazardous winter storms are present, and the surrounding surface near the campus is sparsely populated, the FAA rejected the request without much justification. Nonetheless, leveraging nearby terrain elevation change and structures, the FAA Part 107 rules allowed the aircraft to reach up to 200 m (650 ft.) above ground level.

As discussed in Section 2.1.10, the WxUAS collected thousands of samples per variable, per profile. These samples produced high-resolution snapshots of the atmosphere, allowing researchers to tease out minor details in each variable for each flight. Fig. 15 showed a few visualization examples of the data from the March 9-10, 2024, IOP. Fig. 15a exemplified the cloud droplet size distribution profile where the *y-axis* represents altitude, and the *x-axis* the total particle count for that altitude. In the cloud droplet size distribution profile, the outer color represents the distribution's mode at each altitude, that is, the cloud droplet size bin with the highest count while the inner color represents the largest particle size detected at that altitude. Here, the cloud droplet distribution explained what type of cloud region the WxUAS sampled. Near the surface, there were low particle counts and both the inside and outside coloring were blue, so the WxUAS was in clear air (Fig. 15a);

as the WxUAS approached 100m, the inner color was red and the outer was blue, which indicates a mix of small and large particles see in cloud transition layer, so the WxUAS likely was starting to enter clouds. Fig. 15b presented a micro-range water content which indicates the vertical variations of water content in the cloud droplet size range. Fig. 15c, d represented the vertical profiles for temperature and relative humidity. Finally, Fig. 15e, f highlighted time series for the radar derived approximated hydrometeor fall velocity and average reflectivity.

Fig. 15. Example of data collected from WxUAS during the March 9-10, 2024, IOP deployment. The observations collected included vertical profiles of (a) particle size and count, (b) micro-range water content, (c) temperature, and (d) relative humidity. Additionally, the onboard radar captured information on (e) fall velocity and (f) reflectivity.

# 4 Highlights and Comparisons of the Instrumentation during the February 28th, 2023, Nor'easter

February 28<sup>th</sup> was one of the largest snow events to occur in Southern New England during the winter of 2022-2023. Large portions of Connecticut, Rhode Island, and Western Massachusetts received 6-8 inches of snow with some areas reaching up to 10 inches. This winter storm was forced by a strong mid-level low pressure moving through the Ohio River Valley. There were double surface lows associated with the storm with the main low crossing the Great Lakes and the secondary low developing over the Southern Chesapeake Bay; the southern low-pressure center was closest to UConn as it moved along the coast of the Northeast United States. Significant moisture and forcing was in the region, along with a cold thermodynamic profile, which led to a long duration snow event. The UConn GPM GV sites were in the path of some of the heaviest snow rates, which occurred in the early overnight hours of the 28<sup>th</sup>. Official NWS Boston snow reports showed that observing locations around UConn received 7 inches of snow as of 7am EST (12 UTC).

Despite the GAIL site's location being further west, the first precipitation occurred, as detected by the PARSIVEL<sup>2</sup>, at 01:41 UTC at the D3R site. Precipitation at the GAIL site started slightly later at 02:02 UTC. While one would expect the western site, GAIL, to experience precipitation first due to the storm's general west to east motion, this was not the case due to the higher elevation at the D3R site. There are many possible contributors to this occurring; with the relative humidity was slightly lower at the GAIL site and combining that with a small dry slot aloft, it is likely that the terrain and elevation change of 64 m caused snowflakes to sublimate before reaching the ground. The precipitation lasted until 22:18 UTC at the D3R site and ended shortly after at the GAIL site at 22:37 UTC. This event registered completely as snow to the PARSIVEL<sup>2</sup>'s present weather algorithm at both observing locations.

At the GAIL site, the AIO (Fig. 16a-e) registered significant changes before and during the storm. Fig. 16a highlighted that during the second half of the snow event the air temperature and dewpoint had risen above 0 °C as opposed to being below freezing during the heaviest snowfall; this was important to note because some precipitation phase classifications may misclassify this (e.g., MRMS precipitation phase would classify periods of snow during this event as periods of rain based on NEXRAD radar data, see Zhang et al. 2016 Figure 10). Fig. 16c showed surface pressure values dropping until between 7 and 8 UTC, where they remained relatively constant until after 18 UTC. The wind speed also dropped off significantly during the event after precipitation onset (Fig. 16d, e). The winds were strongest around 8 m s<sup>-1</sup> out of the south before onset of precipitation and dropped to about 3 m s<sup>-1</sup> for most of the day; during the event, the wind shifted from the south to the east to the north as the low-pressure center moved northeast along the coast.

The PARSIVEL<sup>2</sup> and PIP both highlight the heaviest observed precipitation rates occurring between 3 and 7 UTC (Fig. 16g, h). The PARSIVEL<sup>2</sup> snow rate (Fig. 16g) estimates the amount of snowfall accumulation on the ground, hence why it is around a factor of 10 higher than the PIP precipitation rate (Fig. 16h). Most of the precipitation occurred before 13 UTC and became more intermittent as temperatures increased (Fig. 16a). Towards the end of the precipitation, larger snowflakes were forming, according to the maximum and mass-weighted mean diameter (Fig. 16f). This could be an indication of particles aggregating together or riming more as surface temperatures rose. The PIP's equivalent bulk density (eDen) measurements (Fig. 16i) also pointed to this fact (a typical 10:1 snow to liquid ratio appears as 0.1 g cm<sup>-3</sup>); after 16 UTC, eDen values rose to being significantly above 0.1 g cm<sup>-3</sup> indicating heavier, rimed snowflakes. The PSD for both PARSIVEL<sup>2</sup> and PIP (Fig. 16l, m) matched well and highlighted the heavier precipitation at the beginning of the snow event with more numerous larger particles and the largest amounts of the small particles as well. The PARSIVEL<sup>2</sup> tended to observe more small particles than the PIP later in the event, which was important to note when checking validity of the observations. If only one of these instruments was present at GAIL, this difference would not have been noticeable, reinforcing the need for redundant observations.

From a radar perspective, the MRR observations were consistent with the microphysical instrument observations. The highest reflectivity values occurred just after precipitation onset throughout the periods of heaviest precipitation (Fig. 16j), and after 13 UTC, the precipitation was lighter, as previously seen. The Doppler velocities (Fig. 16k) seen by the MRR also matched what would be expected for a snow event; this was because the velocities are slower, typical velocities for snow is 1-3 m s<sup>-1</sup>, and there was no bright band, a sharp gradient in doppler velocities from slow to fast, to indicate a change from ice or snow to rain (Fig. 16j). Greater downward velocities highlighted the heaviest snow rates at the beginning of the event, whereas smaller velocities dominated at low levels as the day progressed.

The MASC and SMAS were able to observe a variety of snowflake types due to the variety of conditions from the heavy snow bands to the lighter, denser snow later in the day. Graupel and aggregates were the most common types of snowflakes seen out of the six available geometric categories for classification (Fig. 16n). In addition, 79.7 % of snowflakes classified had some form of riming during this event (Fig. 16o). The rising temperatures towards the end of the event, visible in Fig. 16a, raised wet-bulb temperatures above freezing, which would induce more riming or particle aggregates as the edges of the ice or snow particles begin to melt increase the water content and density, confirmed by Fig. 16i.

Fig. 16. 5-minute averages of observations collected at GAIL site on February 28, 2023. AIO observations of (a) temperature and dewpoint, (b) relative humidity, (c) surface pressure, (d) wind speed, and (e) wind direction. PARSIVEL<sup>2</sup> observations of (f) maximum and mass-weighted mean diameter, and (g) snow rate. PIP observations of (h) precipitation rate and (i) equivalent bulk density. Reprocessed MRR observations of (j) reflectivity and (k) radial velocity. Particle size distributions for the (l) PARSIVEL<sup>2</sup> and (m) PIP. Geometrical classification of snowflakes (n) using a neural network classifier (Thant et al. 2023b), and a machine learning classification (Thant et al. 2023b) of riming degrees of snowflakes (o) based on all MASC and SMAS images during February 28 event.

At the D3R site, the Pluvio told a similar precipitation story as at the GAIL site. The total accumulation from the Pluvio (Fig. 17a) showed a steady increase in precipitation during the early morning hours matching the high precipitation rates. Later in the day, precipitation rates decreased with less overall accumulating precipitation. The PARSIVEL<sup>2</sup> showed similar trends as its counterpart at the GAIL site. The largest snowflakes at the D3R site occurred during the period of heaviest precipitation at the beginning of the event, and the D3R site tended to have slightly larger flakes than the GAIL Site throughout the second half of the event (Fig. 17b). Despite the larger snowflakes at the D3R site later in the event, the snow rates did not differ significantly from the GAIL site (Fig. 17c). It was not surprising that the two sites return similar observations due to the storm motion.

For the PIP, there were some notable differences between the two sites. The D3R PIP recorded some heavier precipitation rates particularly at the beginning of the event with rates reaching over 5 mm h<sup>-1</sup> (Fig. 17d), whereas the GAIL site's max precipitation rate was around 2.5 mm h<sup>-1</sup> (Fig. 16h). In addition, precipitation rates tended to be greater during the second half of the event due to the D3R site being slightly colder than the GAIL site because of the elevation difference. The colder temperatures may have prevented the snowflakes from beginning to rime or partially melt. This was reflected by the lower eDen values captured by the PIP (Fig. 17e). The PSDs for both the PARSIVEL<sup>2</sup> and PIP (Fig. 17f, g) are very similar during the entire event with the main exception being that the PIP saw more numerous very small and very large particles throughout most of the event.

As for the radar perspective, there were some interesting features visible from the three different radars at the D3R site. For the MRR, there was a similar pattern to what was exhibited at the GAIL site. The reflectivity from the MRR showed the highest values very early in the event (Fig. 17h); it also had some higher values later in the event which were not seen at the GAIL site. The velocity values (Fig. 17i) were also similar to the GAIL site, but they tended to be slightly slower across the entire event. The GAIL had velocities between 2-6 m s<sup>-1</sup> between 1 and 3 km, whereas the D3R site velocity barely reached 3 m s<sup>-1</sup> consistently. The ACHIEVE W-band radar is a nice complement to the MRR as it was generally pointed vertically for the duration of the event and scans to higher altitudes than the MRR. Between 0-9 UTC, the reflectivity peaks from the W-band (Fig. 17j) generally matched the MRR, even if they had differing intensities. Because the W-band transmitter was more powerful, it better highlighted higher cloud structures during the beginning of event, although the W-band signal attenuates more as precipitation rates increase. Fig. 17j clearly shows the descending clouds approaching at the beginning, and it shows more intense periods of reflectivity associated with the timing of heavier precipitation rates. As a complement,

the W-band's LDR (Fig. 17k) indicates departures from spherical particles. During the period of heaviest precipitation, pockets of larger LDR values (light blue shading) are most likely associated with larger aggregates and rimed snow particles.

The D3R radar performed PPI scans, as well as RHI scans over the GAIL site (315° azimuth) and along the ER-2 aircraft track for this snow event. Fig. 17l showed the reflectivity as measured by the Ka band radar on the D3R just after precipitation started. The highest reflectivity values at the lowest height levels indicated the heaviest precipitation was still further away (over 15 km) from the D3R site. The RHI scan NW of the radar showed light reflectivity values near the surface. This correlated with the precipitation not starting at the GAIL site until 2:02 UTC with most of this reflectivity tied to precipitation aloft. Fig. 17m highlighted the doppler velocity from the D3R; this indicated the precipitation was moving away from the D3R (purple and red colors) at the lowest level, which matches the AIO observations recording wind direction from the south southeast. For the most part, the doppler velocities were fairly uniform except at the lowest scanning levels, where there was variation of higher and lower speeds.

Fig. 17. 5-minute averages of observations collected at D3R site on February 28, 2023. Pluvio observations of (a) total accumulated precipitation and precipitation rate. PARSIVEL<sup>2</sup> observations of (b) maximum and mass-weighted mean diameter and (c) snow rate. PIP observations of (d) precipitation rate and (e) equivalent bulk density. Particle size distributions for the (f) PARSIVEL<sup>2</sup> and (g) PIP. Reprocessed MRR observations of (h) reflectivity and (i) radial velocity. ACHIEVE W-band observations of (j) reflectivity and (k) Linear Depolarization Ratio (LDR, ratio of received horizontal to vertical polarization) (Note: Gaps in W-band plots indicate when it performed RHI scans. The horizontal dashed line indicates top of radar blind zone at 450 m.) D3R radar observations of (l) Reflectivity measured by Ka band radar and (m) doppler Velocity, observed at Ku band from RHI scan at 315° azimuth at 01:55 UTC.

### 5 Data Availability

The dataset presented in this article is available through NASA's Global Hydrometeorology Resource Center

(GHRC) at the following campaign wide DOI: <a href="http://dx.doi.org/10.5067/GPMGVUCONN/DATA101">http://dx.doi.org/10.5067/GPMGVUCONN/DATA101</a> (Cerrai et al. 2025).

Table 4 below includes the individual dataset DOIs. All datasets processed by GHRC goes through standardized quality control checks and documentation as certified by the World Data System CoreTrustSeal (Global Hydrometeorology Resource Center 2024). This data processing done at GHRC includes all steps of a dataset's life cycle from ingestion to processing, archiving, cataloguing, and documenting; their work also supports the distribution on final, published datasets.

Accompanying each individual dataset is a user guide that provides additional information and metadata about the dataset including a description of the instrument, file naming conventions, the variables stored within each file, and the data format of the files.

For the external data, the KIJD ASOS observations, used to validate the AIO, can be downloaded from Iowa State University's Environmental Mesonet here: <a href="https://mesonet.agron.iastate.edu/request/asos/1min.phtml">https://mesonet.agron.iastate.edu/request/asos/1min.phtml</a>. The Storrs, CT GHCN site data, used to validate the AIO, can be viewed here: <a href="https://www.nrcc.cornell.edu/wxstation/nowdata.html">https://www.nrcc.cornell.edu/wxstation/nowdata.html</a>.

| Instrument            | DOI                                                                                 |
|-----------------------|-------------------------------------------------------------------------------------|
| MRR                   | http://dx.doi.org/10.5067/GPMGV/UCONN/MRR2/DATA101 (Williams et al. 2025)           |
| Pluvio                | http://dx.doi.org/10.5067/GPMGV/UCONN/PLUVIO/DATA101 (Tokay and Wolff 2025a)        |
| All-in-One-2          | http://dx.doi.org/10.5067/GPMGV/UCONN/AIO2/DATA101 (Filipiak et al. 2025a)          |
| PARSIVEL <sup>2</sup> | http://dx.doi.org/10.5067/GPMGV/UCONN/PARSIVEL/DATA101 (Tokay and Wolff 2025b)      |
| PIP                   | http://dx.doi.org/10.5067/GPMGV/UCONN/PIP/DATA101 (Wolff et al. 2025)               |
| D3R                   | http://dx.doi.org/10.5067/GPMGV/UCONN/D3R/DATA101 (Chandrasekar et al. 2025)        |
| ACHIEVE               | http://dx.doi.org/10.5067/IMPACTS/WBAND/DATA101 (Loftus et al. 2024)                |
| Tipping Buckets       | http://dx.doi.org/10.5067/GPMGV/UCONN/TIPPINGBUCKET/DATA101 (Tokay and Wolff 2025c) |
| SMAS                  | http://dx.doi.org/10.5067/GPMGV/UCONN/SMAS/DATA101 (Thant 2025a)                    |
| MASC                  | http://dx.doi.org/10.5067/GPMGV/UCONN/MASC/DATA101 (Thant 2025b)                    |
| RM Young              | http://dx.doi.org/10.5067/GPMGV/UCONN/ANEMOMETER/DATA101 (Filipiak et al. 2025b)    |
| Ceilometer            | http://dx.doi.org/10.5067/GPMGV/UCONN/CEILO/DATA101 (Loftus 2025)                   |

Table 4. Summary of individual instrument DOIs available at NASA's Global Hydrometeorology Resource Center.

### **6 Conclusions**

These GPM GV campaigns collected high-resolution datasets, with multiple complimentary instruments, across three winters in Connecticut, which produced 117 precipitation events of varying characteristics. These observations captured a wide variety of winter phenomena during their deployment and provided a rich set of data that can be used to conduct research in combination with multiple datasets including other observational field campaigns, numerical weather models, and satellites and other remote sensing instruments. The three years of deployments allowed for variations in instrumentation and site locations, which enhances the quality of observations collected; this diverse collection provides a variety of different ways to combine this observational dataset with other relevant datasets.

A natural pairing of this data is with the GPM satellite datasets as this was the main reason behind the field deployments. Combining satellite products with observations collected at UCONN allows for verification of satellite derived products and evaluation of algorithms in a variety of winter conditions. The wide variety of precipitation phases and weather conditions allow for a substantial evaluation of satellite products in the Northeast United States, where strong and impactful winter storms occur regularly.

Another logical pairing of data is with the IMPACTS field campaign data as there were three intensive observing periods where aircraft passed over or near UCONN sites or mobile sounding teams launched radiosondes near the sites. This dataset presents the opportunity to have a more expanded view of winter storms in the Northeast United States with a broader set of observations. This could include verification of numerical models, comparisons between aircraft, ground-based and remote sensing instruments, which can further refine understanding of winter precipitation processes.

The high-resolution nature of the data collected allows for it to be used in experiments seeking to better understand the physical processes of precipitation. Experiments involving intercomparisons between the ground-based instruments would allow for understanding of how the different types of instrumentation observe the same precipitation. Comparisons between the remote sensing instruments and in-situ instruments are also possible (Billault-Roux et al. 2023; Shates et al. 2025). Other research using this dataset has already been started including improving the PIP's algorithm to accurately identify rain versus snow, refining SWER(ZE) relationships (Tokay et al. 2023; Inglis et al. 2024), used to estimate snow water equivalent rates from radar reflectivity, between the PIP and MRR, understanding microphysical properties of snowfall (Chang et al. 2024; King et al. 2024), and intercomparison of precipitation accumulations from different instrumentation.

Additional high-resolution observations of winter phenomena are an essential part of furthering research to improve our prior knowledge and ability to forecast them. Future work to improve the measurements and understanding of precipitation across the globe relies on high-quality observations, like the ones collected in this NASA GPM GV campaign.

7 Appendix
Appendix A: 2021-2024 Precipitation Event Table

|      |            | Start |            | End   | Precipitation |                    | Total precipitation |
|------|------------|-------|------------|-------|---------------|--------------------|---------------------|
| Site | Start Date | нн:мм | End Date   | нн:мм | Minutes       | <b>Event Phase</b> | (mm)                |
| GAIL | 12/9/2021  | 0:29  | 12/9/2021  | 6:19  | 332           | S                  | 52.17               |
| GAIL | 12/11/2021 | 10:33 | 12/11/2021 | 20:05 | 460           | R                  | 5.99                |
| GAIL | 12/12/2021 | 1:59  | 12/12/2021 | 5:43  | 132           | R                  | 2.79                |
| GAIL | 12/15/2021 | 22:08 | 12/16/2021 | 6:15  | 346           | R                  | 4.93                |
| GAIL | 12/18/2021 | 16:42 | 12/19/2021 | 11:17 | 973           | R                  | 14.26               |
| GAIL | 12/22/2021 | 7:32  | 12/22/2021 | 16:11 | 327           | R                  | 6.97                |
| GAIL | 12/24/2021 | 5:22  | 12/24/2021 | 16:05 | 424           | S                  | 85.6                |
| GAIL | 12/25/2021 | 11:43 | 12/25/2021 | 18:21 | 391           | R                  | 9.06                |
| GAIL | 12/26/2021 | 5:16  | 12/26/2021 | 8:00  | 165           | R                  | 6.47                |
| GAIL | 12/28/2021 | 2:13  | 12/28/2021 | 12:55 | 220           | М                  | 13.11               |
| GAIL | 12/29/2021 | 1:50  | 12/29/2021 | 10:36 | 345           | R                  | 3.04                |
| GAIL | 12/30/2021 | 0:23  | 12/30/2021 | 15:38 | 446           | R                  | 1.15                |
| GAIL | 12/31/2021 | 0:59  | 12/31/2021 | 8:45  | 232           | R                  | 2.88                |
| GAIL | 1/2/2022   | 1:57  | 1/2/2022   | 9:12  | 347           | R                  | 7.74                |
| GAIL | 1/5/2022   | 12:12 | 1/5/2022   | 19:01 | 270           | R                  | 9.78                |
| GAIL | 1/7/2022   | 6:23  | 1/7/2022   | 21:46 | 750           | S                  | 433.78              |
| GAIL | 1/9/2022   | 21:59 | 1/10/2022  | 2:02  | 239           | R                  | 4.37                |
| GAIL | 1/17/2022  | 5:38  | 1/18/2022  | 5:06  | 911           | М                  | 107.17              |
| GAIL | 1/20/2022  | 8:37  | 1/20/2022  | 16:07 | 442           | М                  | 38.12               |
| GAIL | 1/25/2022  | 3:10  | 1/25/2022  | 5:24  | 135           | S                  | 81.81               |

| GAIL | 1/29/2022 | 3:01  | 1/30/2022 | 6:21  | 1434 | S | 391.37 |
|------|-----------|-------|-----------|-------|------|---|--------|
| GAIL | 2/3/2022  | 11:43 | 2/5/2022  | 1:09  | 2026 | R | 48.7   |
| GAIL | 2/7/2022  | 12:51 | 2/8/2022  | 9:30  | 1165 | М | 23.35  |
| GAIL | 2/13/2022 | 8:49  | 2/14/2022 | 2:55  | 686  | S | 137.95 |
| GAIL | 2/18/2022 | 3:26  | 2/18/2022 | 13:10 | 496  | R | 21.54  |
| GAIL | 2/19/2022 | 17:42 | 2/19/2022 | 23:23 | 129  | S | 41.43  |
| GAIL | 2/22/2022 | 17:48 | 2/23/2022 | 3:33  | 450  | R | 26.39  |
| GAIL | 2/25/2022 | 7:22  | 2/25/2022 | 22:17 | 870  | М | 116.2  |
| GAIL | 3/2/2022  | 0:54  | 3/2/2022  | 4:33  | 174  | R | 3.96   |
| GAIL | 3/3/2022  | 6:48  | 3/3/2022  | 9:31  | 158  | R | 6.68   |
| GAIL | 3/6/2022  | 15:26 | 3/6/2022  | 16:28 | 56   | R | 3      |
| GAIL | 3/8/2022  | 3:09  | 3/8/2022  | 3:52  | 23   | R | 2.46   |
| GAIL | 3/9/2022  | 15:47 | 3/10/2022 | 1:17  | 563  | S | 178.06 |
| GAIL | 3/12/2022 | 6:25  | 3/13/2022 | 2:56  | 799  | М | 77.46  |
| GAIL | 3/16/2022 | 2:04  | 3/16/2022 | 3:52  | 109  | R | 1.35   |
| GAIL | 3/17/2022 | 13:05 | 3/17/2022 | 20:47 | 167  | R | 1.16   |
| GAIL | 3/20/2022 | 2:18  | 3/20/2022 | 5:03  | 36   | R | 2.12   |
| GAIL | 3/24/2022 | 1:50  | 3/24/2022 | 19:30 | 932  | R | 22.46  |
| GAIL | 3/25/2022 | 1:27  | 3/25/2022 | 7:58  | 333  | R | 26.76  |
| GAIL | 3/27/2022 | 1:46  | 3/27/2022 | 3:30  | 37   | R | 1.22   |
| GAIL | 4/1/2022  | 0:00  | 4/1/2022  | 12:05 | 481  | R | 31.7   |
| GAIL | 4/3/2022  | 16:08 | 4/4/2022  | 3:40  | 474  | R | 5.09   |
| GAIL | 4/6/2022  | 6:31  | 4/6/2022  | 18:03 | 665  | R | 10.77  |
| GAIL | 4/7/2022  | 20:22 | 4/8/2022  | 13:16 | 901  | R | 33.55  |
| GAIL | 4/9/2022  | 15:50 | 4/9/2022  | 22:19 | 94   | R | 5.03   |
| GAIL | 4/12/2022 | 13:07 | 4/12/2022 | 15:32 | 146  | R | 5.03   |
| GAIL | 4/14/2022 | 23:02 | 4/15/2022 | 3:36  | 107  | R | 4.98   |
| GAIL | 4/16/2022 | 22:35 | 4/17/2022 | 4:04  | 165  | R | 10.03  |

| GAIL         4/19/2022         4:40         4/19/2022         18:22         650         R         48.75           GAIL         4/26/2022         13:40         4/27/2022         5:09         517         R         4.95           GAIL         12/22/2022         20:41         12/23/2022         21:20         1104         R         38.97           D3R         12/22/2022         21:56         12/23/2022         21:19         1053         R         40           GAIL         12/31/2022         18:58         1/1/2023         7:43         620         R         11.71           D3R         12/31/2022         19:00         1/1/2023         7:45         621         R         11.74           D3R         12/31/2023         13:43         1/4/2023         9:48         778         R         16.87           D3R         1/3/2023         13:15         1/4/2023         9:48         778         R         17.21           GAIL         1/4/2023         23:32         1/5/2023         8:55         274         R         3.67           D3R         1/4/2023         23:34         1/5/2023         20:51         555         R         8.25           D                                                                                                                     |      |            |       |            |       |      |   |        |
|---------------------------------------------------------------------------------------------------------------------------------------------------------------------------------------------------------------------------------------------------------------------------------------------------------------------------------------------------------------------------------------------------------------------------------------------------------------------------------------------------------------------------------------------------------------------------------------------------------------------------------------------------------------------------------------------------------------------------------------------------------------------------------------------------------------------------------------------------------------------------------------------------------------------------------------------------------------------------------------------------------------------------------------------------------------------------------------------------------------------------------------------------------------------------------------------------------------------------------------------------------------------------------------------------|------|------------|-------|------------|-------|------|---|--------|
| GAIL         12/22/2022         20:41         12/23/2022         21:10         1104         R         38.97           D3R         12/22/2022         21:56         12/23/2022         21:19         1053         R         40           GAIL         12/31/2022         18:58         1/1/2023         7:43         620         R         11.71           D3R         12/31/2022         19:00         1/1/2023         7:45         621         R         11.48           GAIL         1/3/2023         13:43         1/4/2023         9:47         758         R         16.87           D3R         1/3/2023         13:15         1/4/2023         9:48         778         R         17.21           GAIL         1/4/2023         23:32         1/5/2023         8:55         274         R         3.67           D3R         1/4/2023         23:34         1/5/2023         8:58         272         R         3.26           GAIL         1/6/2023         10:18         1/6/2023         21:03         534         R         7.33           GAIL         1/12/2023         12:21         1/13/2023         15:01         872         R         22.52           D3R                                                                                                                     | GAIL | 4/19/2022  | 4:40  | 4/19/2022  | 18:22 | 650  | R | 48.75  |
| D3R         12/22/2022         21:56         12/23/2022         21:19         1053         R         40           GAIL         12/31/2022         18:58         1/1/2023         7:43         620         R         11.71           D3R         12/31/2022         19:00         1/1/2023         7:45         621         R         11.48           GAIL         1/3/2023         13:43         1/4/2023         9:47         758         R         16.87           D3R         1/3/2023         13:15         1/4/2023         9:48         778         R         17.21           GAIL         1/4/2023         23:32         1/5/2023         8:55         274         R         3.67           D3R         1/4/2023         23:34         1/5/2023         8:58         272         R         3.26           GAIL         1/6/2023         9:55         1/6/2023         20:51         555         R         8.25           D3R         1/12/2023         10:18         1/6/2023         21:03         534         R         7.33           GAIL         1/12/2023         12:21         1/13/2023         14:59         790         R         20.59           D3R                                                                                                                            | GAIL | 4/26/2022  | 13:40 | 4/27/2022  | 5:09  | 517  | R | 4.95   |
| GAIL         12/31/2022         18:58         1/1/2023         7:43         620         R         11.71           D3R         12/31/2022         19:00         1/1/2023         7:45         621         R         11.48           GAIL         1/3/2023         13:43         1/4/2023         9:47         758         R         16.87           D3R         1/3/2023         13:15         1/4/2023         9:48         778         R         17.21           GAIL         1/4/2023         23:32         1/5/2023         8:55         274         R         3.67           D3R         1/4/2023         23:34         1/5/2023         8:58         272         R         3.26           GAIL         1/6/2023         9:55         1/6/2023         20:51         555         R         8.25           D3R         1/6/2023         10:18         1/6/2023         21:03         534         R         7.33           GAIL         1/12/2023         12:21         1/13/2023         15:01         872         R         22.52           D3R         1/14/2023         18:21         1/14/2023         20:22         115         S         5.48           GAIL                                                                                                                             | GAIL | 12/22/2022 | 20:41 | 12/23/2022 | 21:20 | 1104 | R | 38.97  |
| D3R         12/31/2022         19:00         1/1/2023         7:45         621         R         11.48           GAIL         1/3/2023         13:43         1/4/2023         9:47         758         R         16.87           D3R         1/3/2023         13:15         1/4/2023         9:48         778         R         17.21           GAIL         1/4/2023         23:32         1/5/2023         8:55         274         R         3.67           D3R         1/4/2023         23:34         1/5/2023         8:58         272         R         3.26           GAIL         1/6/2023         9:55         1/6/2023         20:51         555         R         8.25           D3R         1/6/2023         10:18         1/6/2023         21:03         534         R         7.33           GAIL         1/12/2023         12:21         1/13/2023         15:01         872         R         22.52           D3R         1/12/2023         12:26         1/13/2023         14:59         790         R         20.59           GAIL         1/14/2023         18:21         1/14/2023         20:22         115         S         5.48           GAIL                                                                                                                            | D3R  | 12/22/2022 | 21:56 | 12/23/2022 | 21:19 | 1053 | R | 40     |
| GAIL         1/3/2023         13:43         1/4/2023         9:47         758         R         16.87           D3R         1/3/2023         13:15         1/4/2023         9:48         778         R         17.21           GAIL         1/4/2023         23:32         1/5/2023         8:55         274         R         3.67           D3R         1/4/2023         23:34         1/5/2023         8:58         272         R         3.26           GAIL         1/6/2023         9:55         1/6/2023         20:51         555         R         8.25           D3R         1/6/2023         10:18         1/6/2023         21:03         534         R         7.33           GAIL         1/12/2023         12:21         1/13/2023         15:01         872         R         22.52           D3R         1/12/2023         12:26         1/13/2023         14:59         790         R         20.59           GAIL         1/14/2023         18:21         1/14/2023         20:22         115         S         5.48           GAIL         1/19/2023         18:05         1/20/2023         5:40         659         R         29.75           D3R                                                                                                                            | GAIL | 12/31/2022 | 18:58 | 1/1/2023   | 7:43  | 620  | R | 11.71  |
| D3R         1/3/2023         13:15         1/4/2023         9:48         778         R         17.21           GAIL         1/4/2023         23:32         1/5/2023         8:55         274         R         3.67           D3R         1/4/2023         23:34         1/5/2023         8:58         272         R         3.26           GAIL         1/6/2023         9:55         1/6/2023         20:51         555         R         8.25           D3R         1/6/2023         10:18         1/6/2023         21:03         534         R         7.33           GAIL         1/12/2023         12:21         1/13/2023         15:01         872         R         22.52           D3R         1/12/2023         12:26         1/13/2023         14:59         790         R         20.59           GAIL         1/14/2023         18:21         1/14/2023         20:22         115         S         5.48           GAIL         1/19/2023         18:05         1/20/2023         5:40         659         R         29.75           D3R         1/19/2023         18:09         1/20/2023         23:53         1451         M         111.27           D3R <td>D3R</td> <td>12/31/2022</td> <td>19:00</td> <td>1/1/2023</td> <td>7:45</td> <td>621</td> <td>R</td> <td>11.48</td> | D3R  | 12/31/2022 | 19:00 | 1/1/2023   | 7:45  | 621  | R | 11.48  |
| GAIL         1/4/2023         23:32         1/5/2023         8:55         274         R         3.67           D3R         1/4/2023         23:34         1/5/2023         8:58         272         R         3.26           GAIL         1/6/2023         9:55         1/6/2023         20:51         555         R         8.25           D3R         1/6/2023         10:18         1/6/2023         21:03         534         R         7.33           GAIL         1/12/2023         12:21         1/13/2023         15:01         872         R         22.52           D3R         1/12/2023         12:26         1/13/2023         14:59         790         R         20.59           GAIL         1/14/2023         18:21         1/14/2023         20:22         115         S         5.48           GAIL         1/19/2023         18:05         1/20/2023         5:40         659         R         29.75           D3R         1/19/2023         18:09         1/20/2023         23:53         1451         M         111.27           D3R         1/122/2023         22:15         1/23/2023         23:48         1424         M         126.49                                                                                                                                | GAIL | 1/3/2023   | 13:43 | 1/4/2023   | 9:47  | 758  | R | 16.87  |
| D3R         1/4/2023         23:34         1/5/2023         8:58         272         R         3.26           GAIL         1/6/2023         9:55         1/6/2023         20:51         555         R         8.25           D3R         1/6/2023         10:18         1/6/2023         21:03         534         R         7.33           GAIL         1/12/2023         12:21         1/13/2023         15:01         872         R         22.52           D3R         1/12/2023         12:26         1/13/2023         14:59         790         R         20.59           GAIL         1/14/2023         18:21         1/14/2023         20:22         115         S         5.48           GAIL         1/19/2023         18:05         1/20/2023         5:40         659         R         29.75           D3R         1/19/2023         18:09         1/20/2023         5:40         654         R         30.08           GAIL         1/22/2023         22:15         1/23/2023         23:53         1451         M         111.27           D3R         1/125/2023         20:36         1/25/2023         21:33         57         S         5.71           G                                                                                                                     | D3R  | 1/3/2023   | 13:15 | 1/4/2023   | 9:48  | 778  | R | 17.21  |
| GAIL         1/6/2023         9:55         1/6/2023         20:51         555         R         8.25           D3R         1/6/2023         10:18         1/6/2023         21:03         534         R         7.33           GAIL         1/12/2023         12:21         1/13/2023         15:01         872         R         22.52           D3R         1/12/2023         12:26         1/13/2023         14:59         790         R         20.59           GAIL         1/14/2023         18:21         1/14/2023         20:22         115         S         5.48           GAIL         1/19/2023         18:05         1/20/2023         5:40         659         R         29.75           D3R         1/19/2023         18:09         1/20/2023         23:53         1451         M         111.27           D3R         1/19/2023         22:15         1/23/2023         23:53         1451         M         111.27           D3R         1/22/2023         22:22         1/23/2023         23:48         1424         M         126.49           GAIL         1/25/2023         23:55         1/26/2023         10:38         642         R         38.59                                                                                                                       | GAIL | 1/4/2023   | 23:32 | 1/5/2023   | 8:55  | 274  | R | 3.67   |
| D3R         1/6/2023         10:18         1/6/2023         21:03         534         R         7.33           GAIL         1/12/2023         12:21         1/13/2023         15:01         872         R         22.52           D3R         1/12/2023         12:26         1/13/2023         14:59         790         R         20.59           GAIL         1/14/2023         18:21         1/14/2023         20:22         115         S         5.48           GAIL         1/19/2023         18:05         1/20/2023         5:40         659         R         29.75           D3R         1/19/2023         18:09         1/20/2023         5:40         654         R         30.08           GAIL         1/22/2023         22:15         1/23/2023         23:53         1451         M         111.27           D3R         1/22/2023         22:22         1/23/2023         23:48         1424         M         126.49           GAIL         1/25/2023         20:36         1/25/2023         21:33         57         S         5.71           GAIL         1/25/2023         20:38         1/26/2023         10:38         642         R         38.59                                                                                                                       | D3R  | 1/4/2023   | 23:34 | 1/5/2023   | 8:58  | 272  | R | 3.26   |
| GAIL         1/12/2023         12:21         1/13/2023         15:01         872         R         22:52           D3R         1/12/2023         12:26         1/13/2023         14:59         790         R         20:59           GAIL         1/14/2023         18:21         1/14/2023         20:22         115         S         5.48           GAIL         1/19/2023         18:05         1/20/2023         5:40         659         R         29:75           D3R         1/19/2023         18:09         1/20/2023         5:40         654         R         30:08           GAIL         1/22/2023         22:15         1/23/2023         23:53         1451         M         111.27           D3R         1/22/2023         22:22         1/23/2023         23:48         1424         M         126:49           GAIL         1/25/2023         20:36         1/25/2023         21:33         57         S         5.71           GAIL         1/25/2023         20:38         1/25/2023         21:33         55         S         4.61           D3R         1/25/2023         23:56         1/26/2023         10:40         644         R         34.46                                                                                                                      | GAIL | 1/6/2023   | 9:55  | 1/6/2023   | 20:51 | 555  | R | 8.25   |
| D3R         1/12/2023         12:26         1/13/2023         14:59         790         R         20.59           GAIL         1/14/2023         18:21         1/14/2023         20:22         115         S         5.48           GAIL         1/19/2023         18:05         1/20/2023         5:40         659         R         29.75           D3R         1/19/2023         18:09         1/20/2023         5:40         654         R         30.08           GAIL         1/22/2023         22:15         1/23/2023         23:53         1451         M         111.27           D3R         1/22/2023         22:22         1/23/2023         23:48         1424         M         126.49           GAIL         1/25/2023         20:36         1/25/2023         21:33         57         S         5.71           GAIL         1/25/2023         23:55         1/26/2023         10:38         642         R         38.59           D3R         1/25/2023         23:56         1/26/2023         10:40         644         R         34.46           GAIL         1/31/2023         9:22         1/31/2023         13:10         122         S         3.04                                                                                                                      | D3R  | 1/6/2023   | 10:18 | 1/6/2023   | 21:03 | 534  | R | 7.33   |
| GAIL         1/14/2023         18:21         1/14/2023         20:22         115         S         5.48           GAIL         1/19/2023         18:05         1/20/2023         5:40         659         R         29.75           D3R         1/19/2023         18:09         1/20/2023         5:40         654         R         30.08           GAIL         1/22/2023         22:15         1/23/2023         23:53         1451         M         111.27           D3R         1/22/2023         22:22         1/23/2023         23:48         1424         M         126.49           GAIL         1/25/2023         20:36         1/25/2023         21:33         57         S         5.71           GAIL         1/25/2023         23:55         1/26/2023         10:38         642         R         38.59           D3R         1/25/2023         20:38         1/25/2023         21:33         55         S         4.61           D3R         1/25/2023         23:56         1/26/2023         10:40         644         R         34.46           GAIL         1/31/2023         9:22         1/31/2023         13:10         122         S         3.04                                                                                                                        | GAIL | 1/12/2023  | 12:21 | 1/13/2023  | 15:01 | 872  | R | 22.52  |
| GAIL         1/19/2023         18:05         1/20/2023         5:40         659         R         29.75           D3R         1/19/2023         18:09         1/20/2023         5:40         654         R         30.08           GAIL         1/22/2023         22:15         1/23/2023         23:53         1451         M         111.27           D3R         1/22/2023         22:22         1/23/2023         23:48         1424         M         126.49           GAIL         1/25/2023         20:36         1/25/2023         21:33         57         S         5.71           GAIL         1/25/2023         23:55         1/26/2023         10:38         642         R         38.59           D3R         1/25/2023         20:38         1/25/2023         21:33         55         S         4.61           D3R         1/25/2023         23:56         1/26/2023         10:40         644         R         34.46           GAIL         1/31/2023         9:22         1/31/2023         13:10         122         S         3.04           D3R         1/31/2023         1:56         2/8/2023         3:57         97         R         3.94           <                                                                                                                 | D3R  | 1/12/2023  | 12:26 | 1/13/2023  | 14:59 | 790  | R | 20.59  |
| D3R         1/19/2023         18:09         1/20/2023         5:40         654         R         30.08           GAIL         1/22/2023         22:15         1/23/2023         23:53         1451         M         111.27           D3R         1/22/2023         22:22         1/23/2023         23:48         1424         M         126.49           GAIL         1/25/2023         20:36         1/25/2023         21:33         57         S         5.71           GAIL         1/25/2023         23:55         1/26/2023         10:38         642         R         38.59           D3R         1/25/2023         20:38         1/25/2023         21:33         55         S         4.61           D3R         1/25/2023         23:56         1/26/2023         10:40         644         R         34.46           GAIL         1/31/2023         9:22         1/31/2023         13:10         122         S         3.04           D3R         1/31/2023         9:22         1/31/2023         13:12         155         S         4.72           GAIL         2/8/2023         1:56         2/8/2023         3:57         96         R         3.84                                                                                                                               | GAIL | 1/14/2023  | 18:21 | 1/14/2023  | 20:22 | 115  | S | 5.48   |
| GAIL         1/22/2023         22:15         1/23/2023         23:53         1451         M         111.27           D3R         1/22/2023         22:22         1/23/2023         23:48         1424         M         126.49           GAIL         1/25/2023         20:36         1/25/2023         21:33         57         S         5.71           GAIL         1/25/2023         23:55         1/26/2023         10:38         642         R         38.59           D3R         1/25/2023         20:38         1/25/2023         21:33         55         S         4.61           D3R         1/25/2023         23:56         1/26/2023         10:40         644         R         34.46           GAIL         1/31/2023         9:22         1/31/2023         13:10         122         S         3.04           D3R         1/31/2023         9:22         1/31/2023         13:12         155         S         4.72           GAIL         2/8/2023         1:56         2/8/2023         3:57         97         R         3.94           D3R         2/8/2023         1:37         2/8/2023         3:57         96         R         3.84                                                                                                                                    | GAIL | 1/19/2023  | 18:05 | 1/20/2023  | 5:40  | 659  | R | 29.75  |
| D3R         1/22/2023         22:22         1/23/2023         23:48         1424         M         126.49           GAIL         1/25/2023         20:36         1/25/2023         21:33         57         S         5.71           GAIL         1/25/2023         23:55         1/26/2023         10:38         642         R         38.59           D3R         1/25/2023         20:38         1/25/2023         21:33         55         S         4.61           D3R         1/25/2023         23:56         1/26/2023         10:40         644         R         34.46           GAIL         1/31/2023         9:22         1/31/2023         13:10         122         S         3.04           D3R         1/31/2023         9:22         1/31/2023         13:12         155         S         4.72           GAIL         2/8/2023         1:56         2/8/2023         3:57         97         R         3.94           D3R         2/8/2023         1:37         2/8/2023         3:57         96         R         3.84                                                                                                                                                                                                                                                         | D3R  | 1/19/2023  | 18:09 | 1/20/2023  | 5:40  | 654  | R | 30.08  |
| GAIL         1/25/2023         20:36         1/25/2023         21:33         57         S         5.71           GAIL         1/25/2023         23:55         1/26/2023         10:38         642         R         38.59           D3R         1/25/2023         20:38         1/25/2023         21:33         55         S         4.61           D3R         1/25/2023         23:56         1/26/2023         10:40         644         R         34.46           GAIL         1/31/2023         9:22         1/31/2023         13:10         122         S         3.04           D3R         1/31/2023         9:22         1/31/2023         13:12         155         S         4.72           GAIL         2/8/2023         1:56         2/8/2023         3:57         97         R         3.94           D3R         2/8/2023         1:37         2/8/2023         3:57         96         R         3.84                                                                                                                                                                                                                                                                                                                                                                             | GAIL | 1/22/2023  | 22:15 | 1/23/2023  | 23:53 | 1451 | М | 111.27 |
| GAIL       1/25/2023       23:55       1/26/2023       10:38       642       R       38.59         D3R       1/25/2023       20:38       1/25/2023       21:33       55       S       4.61         D3R       1/25/2023       23:56       1/26/2023       10:40       644       R       34.46         GAIL       1/31/2023       9:22       1/31/2023       13:10       122       S       3.04         D3R       1/31/2023       9:22       1/31/2023       13:12       155       S       4.72         GAIL       2/8/2023       1:56       2/8/2023       3:57       97       R       3.94         D3R       2/8/2023       1:37       2/8/2023       3:57       96       R       3.84                                                                                                                                                                                                                                                                                                                                                                                                                                                                                                                                                                                                            | D3R  | 1/22/2023  | 22:22 | 1/23/2023  | 23:48 | 1424 | М | 126.49 |
| D3R       1/25/2023       20:38       1/25/2023       21:33       55       S       4.61         D3R       1/25/2023       23:56       1/26/2023       10:40       644       R       34.46         GAIL       1/31/2023       9:22       1/31/2023       13:10       122       S       3.04         D3R       1/31/2023       9:22       1/31/2023       13:12       155       S       4.72         GAIL       2/8/2023       1:56       2/8/2023       3:57       97       R       3.94         D3R       2/8/2023       1:37       2/8/2023       3:57       96       R       3.84                                                                                                                                                                                                                                                                                                                                                                                                                                                                                                                                                                                                                                                                                                               | GAIL | 1/25/2023  | 20:36 | 1/25/2023  | 21:33 | 57   | S | 5.71   |
| D3R       1/25/2023       23:56       1/26/2023       10:40       644       R       34.46         GAIL       1/31/2023       9:22       1/31/2023       13:10       122       S       3.04         D3R       1/31/2023       9:22       1/31/2023       13:12       155       S       4.72         GAIL       2/8/2023       1:56       2/8/2023       3:57       97       R       3.94         D3R       2/8/2023       1:37       2/8/2023       3:57       96       R       3.84                                                                                                                                                                                                                                                                                                                                                                                                                                                                                                                                                                                                                                                                                                                                                                                                               | GAIL | 1/25/2023  | 23:55 | 1/26/2023  | 10:38 | 642  | R | 38.59  |
| GAIL       1/31/2023       9:22       1/31/2023       13:10       122       S       3.04         D3R       1/31/2023       9:22       1/31/2023       13:12       155       S       4.72         GAIL       2/8/2023       1:56       2/8/2023       3:57       97       R       3.94         D3R       2/8/2023       1:37       2/8/2023       3:57       96       R       3.84                                                                                                                                                                                                                                                                                                                                                                                                                                                                                                                                                                                                                                                                                                                                                                                                                                                                                                                 | D3R  | 1/25/2023  | 20:38 | 1/25/2023  | 21:33 | 55   | S | 4.61   |
| D3R       1/31/2023       9:22       1/31/2023       13:12       155       S       4.72         GAIL       2/8/2023       1:56       2/8/2023       3:57       97       R       3.94         D3R       2/8/2023       1:37       2/8/2023       3:57       96       R       3.84                                                                                                                                                                                                                                                                                                                                                                                                                                                                                                                                                                                                                                                                                                                                                                                                                                                                                                                                                                                                                  | D3R  | 1/25/2023  | 23:56 | 1/26/2023  | 10:40 | 644  | R | 34.46  |
| GAIL     2/8/2023     1:56     2/8/2023     3:57     97     R     3.94       D3R     2/8/2023     1:37     2/8/2023     3:57     96     R     3.84                                                                                                                                                                                                                                                                                                                                                                                                                                                                                                                                                                                                                                                                                                                                                                                                                                                                                                                                                                                                                                                                                                                                                | GAIL | 1/31/2023  | 9:22  | 1/31/2023  | 13:10 | 122  | S | 3.04   |
| D3R         2/8/2023         1:37         2/8/2023         3:57         96         R         3.84                                                                                                                                                                                                                                                                                                                                                                                                                                                                                                                                                                                                                                                                                                                                                                                                                                                                                                                                                                                                                                                                                                                                                                                                 | D3R  | 1/31/2023  | 9:22  | 1/31/2023  | 13:12 | 155  | S | 4.72   |
|                                                                                                                                                                                                                                                                                                                                                                                                                                                                                                                                                                                                                                                                                                                                                                                                                                                                                                                                                                                                                                                                                                                                                                                                                                                                                                   | GAIL | 2/8/2023   | 1:56  | 2/8/2023   | 3:57  | 97   | R | 3.94   |
| CAIL 2/17/2022 14:15 2/17/2022 22:11 224 B                                                                                                                                                                                                                                                                                                                                                                                                                                                                                                                                                                                                                                                                                                                                                                                                                                                                                                                                                                                                                                                                                                                                                                                                                                                        | D3R  | 2/8/2023   | 1:37  | 2/8/2023   | 3:57  | 96   | R | 3.84   |
| GAIL 2/1//2023   14.15   2/1//2023   23.11   234   K   4.2                                                                                                                                                                                                                                                                                                                                                                                                                                                                                                                                                                                                                                                                                                                                                                                                                                                                                                                                                                                                                                                                                                                                                                                                                                        | GAIL | 2/17/2023  | 14:15 | 2/17/2023  | 23:11 | 234  | R | 4.2    |

| D3R  | 2/17/2023  | 14:16 | 2/17/2023  | 23:13 | 244  | R | 4.32   |
|------|------------|-------|------------|-------|------|---|--------|
| GAIL | 2/21/2023  | 8:42  | 2/21/2023  | 13:10 | 253  | S | 16.76  |
| D3R  | 2/21/2023  | 8:46  | 2/21/2023  | 13:11 | 244  | S | 35.96  |
| GAIL | 2/22/2023  | 21:12 | 2/23/2023  | 15:17 | 683  | М | 11.4   |
| D3R  | 2/22/2023  | 21:10 | 2/23/2023  | 16:27 | 721  | М | 14.87  |
| GAIL | 2/25/2023  | 16:13 | 2/25/2023  | 22:50 | 160  | S | 14.61  |
| D3R  | 2/25/2023  | 16:08 | 2/25/2023  | 23:25 | 175  | S | 15.66  |
| GAIL | 2/28/2023  | 2:02  | 2/28/2023  | 22:37 | 1178 | S | 211    |
| D3R  | 2/28/2023  | 1:41  | 2/28/2023  | 22:18 | 1191 | S | 199.41 |
| GAIL | 3/2/2023   | 7:42  | 3/2/2023   | 13:57 | 313  | R | 4.42   |
| D3R  | 3/2/2023   | 7:45  | 3/2/2023   | 13:57 | 305  | R | 4.41   |
| GAIL | 3/4/2023   | 1:32  | 3/4/2023   | 18:33 | 966  | М | 82.8   |
| D3R  | 3/4/2023   | 8:52  | 3/4/2023   | 18:10 | 968  | М | 75.75  |
| GAIL | 3/11/2023  | 3:42  | 3/11/2023  | 14:37 | 653  | М | 12.47  |
| D3R  | 3/11/2023  | 3:39  | 3/11/2023  | 14:54 | 647  | М | 36.78  |
| GAIL | 3/13/2023  | 17:38 | 3/15/2023  | 13:55 | 2343 | М | 172.44 |
| D3R  | 3/13/2023  | 17:55 | 3/15/2023  | 13:15 | 2454 | М | 219.46 |
| GAIL | 3/23/2023  | 16:21 | 3/23/2023  | 19:20 | 120  | R | 3.37   |
| D3R  | 3/23/2023  | 16:24 | 3/23/2023  | 19:22 | 121  | R | 3.53   |
| GAIL | 3/25/2023  | 16:29 | 3/26/2023  | 2:14  | 501  | R | 3.67   |
| D3R  | 3/25/2023  | 16:14 | 3/26/2023  | 2:03  | 466  | R | 3.19   |
| GAIL | 3/27/2023  | 21:39 | 3/28/2023  | 16:07 | 728  | R | 7.49   |
| D3R  | 3/27/2023  | 22:08 | 3/28/2023  | 16:05 | 764  | R | 8.77   |
| GAIL | 3/30/2023  | 5:00  | 3/30/2023  | 6:37  | 77   | М | 15.57  |
| D3R  | 3/30/2023  | 5:00  | 3/30/2023  | 6:37  | 58   | М | 10.52  |
| GAIL | 4/1/2023   | 9:41  | 4/1/2023   | 16:12 | 308  | R | 9.3    |
| D3R  | 4/1/2023   | 9:43  | 4/1/2023   | 16:15 | 310  | R | 9.7    |
| GAIL | 12/17/2023 | 20:22 | 12/18/2023 | 20:31 | 1163 | R | 82.36  |
| •    |            | •     | •          | •     | •    | • | •      |

| GAIL | 12/28/2023 | 4:16  | 12/28/2023 | 15:34 | 663  | R | 30.89  |
|------|------------|-------|------------|-------|------|---|--------|
| GAIL | 12/29/2023 | 1:14  | 12/29/2023 | 20:42 | 117  | R | 1.05   |
| GAIL | 1/7/2024   | 0:36  | 1/7/2024   | 23:21 | 1285 | M | 212.57 |
|      | 1/9/2024   |       |            |       |      | R |        |
| GAIL |            | 18:33 | 1/10/2024  | 11:17 | 885  |   | 67.53  |
| GAIL | 1/13/2024  | 5:10  | 1/13/2024  | 16:13 | 530  | R | 27.14  |
| GAIL | 1/16/2024  | 5:36  | 1/16/2024  | 23:43 | 864  | М | 45.51  |
| GAIL | 1/19/2024  | 15:51 | 1/19/2024  | 21:59 | 275  | М | 9.75   |
| GAIL | 1/23/2024  | 18:48 | 1/25/2024  | 12:58 | 1492 | S | 65.7   |
| GAIL | 1/26/2024  | 6:35  | 1/26/2024  | 16:02 | 420  | R | 11.21  |
| GAIL | 1/28/2024  | 8:40  | 1/29/2024  | 6:55  | 1258 | S | 45.6   |
| GAIL | 2/13/2024  | 9:18  | 2/13/2024  | 19:27 | 530  | М | 483.88 |
| GAIL | 2/16/2024  | 1:13  | 2/16/2024  | 2:03  | 50   | М | 9.13   |
| GAIL | 2/17/2024  | 7:43  | 2/17/2024  | 16:10 | 315  | М | 16.27  |
| GAIL | 2/23/2024  | 0:21  | 2/23/2024  | 11:16 | 421  | R | 4.49   |
| GAIL | 2/28/2024  | 1:16  | 2/28/2024  | 23:53 | 967  | R | 17.17  |
| GAIL | 2/29/2024  | 2:16  | 2/29/2024  | 6:02  | 159  | S | 7.92   |
| GAIL | 3/2/2024   | 14:44 | 3/3/2024   | 5:58  | 845  | R | 28.45  |
| GAIL | 3/5/2024   | 13:39 | 3/6/2024   | 1:05  | 416  | R | 8.37   |
| GAIL | 3/6/2024   | 19:09 | 3/7/2024   | 16:14 | 1169 | R | 46.29  |
| GAIL | 3/9/2024   | 21:07 | 3/10/2024  | 9:28  | 610  | R | 33.09  |
| GAIL | 3/11/2024  | 5:07  | 3/11/2024  | 11:04 | 197  | М | 22.86  |
| GAIL | 3/17/2024  | 9:33  | 3/17/2024  | 12:26 | 90   | R | 1.5    |
| GAIL | 3/20/2024  | 21:30 | 3/21/2024  | 0:43  | 162  | S | 2.39   |
| GAIL | 3/23/2024  | 5:01  | 3/24/2024  | 0:23  | 1022 | R | 63.82  |
| GAIL | 3/27/2024  | 20:53 | 3/29/2024  | 9:09  | 1479 | R | 35.44  |
| GAIL | 4/2/2024   | 19:09 | 4/3/2024   | 2:54  | 462  | R | 7.96   |
| GAIL | 4/3/2024   | 11:41 | 4/4/2024   | 18:17 | 1448 | S | 30.92  |
| GAIL | 4/11/2024  | 12:04 | 4/12/2024  | 19:56 | 890  | R | 28.37  |

| GAIL   | 4/17/2024            | 23:49 | 4/18/2024 | 21:03 | 794  | R  | 18.37 |
|--------|----------------------|-------|-----------|-------|------|----|-------|
| OAIL   | 4/1//2024            | 20.40 | 4/10/2024 | 21.00 | 754  | 11 | 10.07 |
| GAIL   | 4/20/2024            | 7:48  | 4/20/2024 | 14:18 | 234  | R  | 5.83  |
| CAII   | 4/00/0004            | 1.40  | 4/00/0004 | 0.00  | 101  | D  | 4.07  |
| GAIL   | 4/28/2024            | 1:43  | 4/28/2024 | 8:09  | 181  | R  | 4.07  |
| GAIL   | 4/28/2024            | 21:40 | 4/29/2024 | 3:15  | 155  | R  | 2.83  |
|        |                      |       |           |       |      |    |       |
| GAIL   | 5/1/2024             | 2:10  | 5/1/2024  | 9:05  | 236  | R  | 3.14  |
| GAIL   | 5/5/2024             | 18:05 | 5/6/2024  | 7:28  | 650  | R  | 24.14 |
|        |                      |       |           |       |      |    |       |
| GAIL   | 5/8/2024             | 11:48 | 5/8/2024  | 13:43 | 115  | R  | 15.02 |
| GAIL   | 5/9/2024             | 0:45  | 5/9/2024  | 0:56  | 12   | R  | 4.05  |
| UAIL   | 3/3/2024             | 0.43  | 3/3/2024  | 0.50  | 12   | 11 | 4.03  |
| GAIL   | 5/10/2024            | 0:45  | 5/10/2024 | 11:49 | 320  | R  | 2.44  |
|        |                      |       |           |       |      |    |       |
| GAIL   | 5/15/2024            | 14:01 | 5/16/2024 | 16:50 | 1473 | R  | 45.3  |
| GAIL   | 5/18/2024            | 10:12 | 5/19/2024 | 2:18  | 597  | R  | 9.9   |
| J. 112 | 5. 25. 2 <b>02</b> 1 |       | 0.20.2021 |       |      | •• |       |

Table A1. Summary table of precipitation events during 2021-2024. Included are the site, start and end time of precipitation, in UTC, the number of precipitating minutes during the event, the precipitation phase during the event, and total precipitation in mm. For precipitation phase, rain events are labeled as R; snow events are labeled S; mixed phase or transition events are labeled as M.

#### **8 Author Contributions**

BF wrote the original first draft, conducted some of the analysis, and produced most of the visualizations. DW and DC were responsible for the conceptualization, supervision, and funding of the field campaigns. AT, CH, AL, LB, VC, and GA provided input on the instrumentation used and deployed during the field campaigns. BF, AS, AC, CS, MB, EK, FJ, HT, and GA were responsible for installing and maintaining the instrumentation. AT, CP, AL, EK, HT, GA assisted with the development and production of some of the visualizations and analysis. BF, DW, AT, CH, AL, CP, VC, BN, GH, and DC all contributed to the writing and editing of the manuscript.

# **9 Competing Interests**

The authors declare that they have no conflict of interest.

#### 10 Acknowledgements

This field campaign was funded by NASA grant 80NSSC20K0577. We would like to thank the UConn Facilities staff for all their assistance in preparing the sites and helping maintain operations throughout the duration of the campaign, and

NASA's Global Hydrometeorology Resource Center staff for their assistance with archiving the data. We would also like to thank Dr. George Huffman, GPM Project Scientist, and Dr. Will McCarty, GPM Program Scientist, for their support. The participation of V. Chandrasekar, EunYeol Kim, and Francesc Junyent was supported by NASA grant (80NSSC19K0877).

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
