# Peer review of "Winter Precipitation Measurements in New England: Results from the Global Precipitation Measurement Ground Validation Campaign in Connecticut"

_Earth System Science Data, 2025_

## Author Comment (AC1)

Responses to Referee 1 Comments for ESSD-2025-162: Winter Precipitation Measurements in New England: Results from the Global Precipitation Measurement Ground Validation Campaign in Connecticut

**Referee Comments 1 (RC1):**

**Overall Comments:**

The paper by Filipiak et al., provides a detailed overview of winter precipitation instruments and derived data products from the multi-year observation sites at the University of Connecticut (UConn). Spanning 3 years, there were 117 precipitation events observed across a collection of 40 instruments, providing a detailed suite of surface and atmospheric variables for tracking the evolving state of the falling particles and meteorological conditions across multiple seasons. The multiple instrument redundancies, data QA, and extensive observational sample results in a high quality dataset that can enhance spaceborne retrievals and model parameterizations in future studies. I find many papers often forget to focus on the importance of good, robust datasets, and am therefore excited to see more data papers like this for solid and mixed-phase precipitation being released. I feel that after the authors address a few minor comments and questions below this paper will be in an acceptable state for publication in ESSD, and will be of great interest to its general readership.

We thank the reviewer for their feedback. We have addressed all their comments below. Please note that the lines mentioned in the responses refer to the clean copy of the revised manuscript.

**General Comments:**

1. **Application Comparison:** I really liked the final sections of the paper demonstrating the consistency across multiple instruments (e.g., Section 4, Figs. 16 and 17), and how this combination of observations provides a level of certainty in individual product accuracy, along with a more holistic view of the meteorologic conditions at the time of observation. However, as is touted early on in the paper and in the conclusions section, one of the major motivators for this work is to help enhance the GPM algorithm for mixed-phase and solid precipitation. I feel it would be of interest to the reader to also include a very brief comparison between these surface measurements/upward pointing radar in this section to DPR measurements from nearby GPM overpasses (e.g., 20230228-S065400-E082628 in the morning and 20230228-S221849-E235117 in the afternoon). This would go a long way in demonstrating the connection between surface and spaceborne and help motivate the applications of this work even more clearly to the reader if GPM was able to provide some spaceborne insights into the same storm.

   1. Thank you for your comment. While we agree that using the data from an overpass from the GPM core observatory would highlight the connection and ability of the instruments to be used in remote sensing applications, we feel that this would deviate too much from the journal's scope, which is focused on the dataset and the methods/instrumentation used to create/collect it, and this paper's focus, which is about the instrumentation and methodology behind the observations collected during the field deployment. Our case study event in Section 4 is intended to showcase the instruments and how they can be used in a collaborative manner rather than comparing or evaluating against other datasets. In doing this, we are leaving these topics to be explored in future research, which can pull from multiple sources of information including this dataset and other datasets as necessary.

2. **Figure Improvements:** While I feel that many of the photos early on were excellent, for a paper that is visually comparing the results of multiple instruments, I think some of the other figures could be improved for visual clarity. For instance:

   1. **Figure 1:** Increasing the size of the site markers and labels would make this easier to read (also missing space after 'Elevation' and 'Distance' in panel b axis labels).

      - We have made these changes to increase the size and the text. See the updated figure on line 140 and below.

[Figure]

2. **Figure 14:** The font is quite small on both panels here making it challenging to read. I would recommend increasing this by quite a bit and making the plot lines thicker.

   - We have made these changes to increase the size of the text and the line thickness. See the updated figure on line 456 and below.

[Figure]

3. **Figure 15:** I like this figure, but I would make panel (b) use the same font size as the others (larger). Additionally, it might be easier to compare differences in the panel (a) colorbar if a perceptually uniform cmap is used instead of jet.

- We have made these changes to this figure including changes to figure layout and size. Additionally, we tried to change the color scheme from jet, but changing to a different color scheme made the differences between the inner and outer colors difficult to interpret in panel a. See the updated figure on line 506 or below.

[Figure]

4. **Figure 16:** This is a great figure that shows quite a bit of relevant information, however the labels are too small to read and have different font sizes. I would recommend getting everything to the same scale and larger in general. For subpanel (j) I would also use a non-discrete perceptually uniform cmap and in panel (k) I'd recommend using bwr centered at zero. For (n) and (o), I would make the pie chart labels white so they are more visible. Out of curiosity, in panel (g) the snow rate is ~10 times that of the non-rain precipitation rate reported in panel (h), is this correct?

- We have made these changes to increase the text size, line weights, label colors, and overall color schemes. For the color schemes selected, we have moved to unified, continuous color schemes that that we believe best represent the data in a way where all features are visible and accessible to all. See the updated figure on line 563 or below.

- For the snowfall rate in panel g, this is correct. The Parsivel infers a snowfall rate in which it is the rate of accumulating snow, not snow water equivalent rate, so this is why the difference is generally around a factor of ~10. We have added additional text to clarify this on lines 538-539 and below

    1. "The PARSIVEL$^2$ snow rate (Fig. 16g) estimates the amount of snowfall accumulation on the ground, hence why it is around a factor of 10 higher than the PIP precipitation rate (Fig. 16h)."

[Figure]

5. **Figure 17:** Similar to my comments for Figure 16, but also I would add the variable and units to the colorbar for panels (h-m) instead of squeezing everything into the title.

- We have made these changes to increase the text size, line weights, text locations, and overall color schemes. For the color schemes selected, we have moved to unified, continuous color schemes that are commonly used. For the color schemes selected, we have moved to unified, continuous color schemes that that we believe best represent the data in a way where all features are visible and accessible to all. See the updated figure on line 609 or below.

[Figure]

**Specific Comments:**

1. **Lines 83-85:** While GPM is certainly better equipped to capture lighter precipitation and falling snow to TRMM, the way this is worded doesn't make it clear that GPM isn't great at snowfall and really underestimates light snowfall in particular (e.g., Casella et al., 2017). I would maybe rework this sentence to make this clearer to the reader.

    1. We have clarified this sentence on lines 83-86, and added additional references if readers are interested. See specific text below as well.

"GPM presented an improvement over its predecessor, the Tropical Rainfall Measurement Mission (TRMM), because of its expansion to include the mid latitude region and ability to detect light precipitation rates and snowfall, albeit not without its challenges (Casella et al. 2017; Milani and Kidd 2023)."

2. **Lines 90-91:** Keep an eye out for the S2noCLIME (Snow Sensitivity to Clouds in a Mountain Environment) data in the coming months which uses many of these same instruments.

    1. We are looking forward to seeing the data collected and possible overlap and collaborative research topics between these field campaigns.

3. **Lines 162-164:** You may want to mention that the two near surface bins from the MRR-2 are typically not used due to surface clutter issues (unless this is addressed here in some form?).

    1. We have made note of this fact on line 166. See specific text below as well.

"During its deployment, the vertical resolution was set to 35 m and had a 30-second raw sampling frequency which was averaged to 1-minute, which allows for observations of the lowest 2 km of the atmosphere (the two range gates nearest the surface bins are not typically used due to potential ground clutter issues). Raw MRR-2 observations collected were for reflectivity, doppler velocity, and spectrum width with additional processing creating additional quantities such as rain rate and liquid water content."

4. **Line 189:** It might be nice to include some general statistics around undercatch even with shields as described in Smith 2007, or Pierre et al., 2019, since I am assuming no additional transfer functions were used?

    1. It is true that no additional transfer functions were applied. However, it would be difficult for us to generate statistics for this due to changes in instrumentation type and locations across the three different winters. Additionally, we did not have a DFIR setup, which both of the studies used as the reference; if we did pursue this, we would have to use the Pluvio as the reference, but then, we could not quantify the impact of undercatch because the windshields were only on the Pluvios . We have done some research comparing how the different precipitation gauges observe both liquid and frozen precipitation, but this is ongoing and out of scope for this paper.

5. **Lines 219-220:** It is great to see the multiple redundant instruments being used for additional quality control!

    1. Thank you for this comment. We hope the data collected between the same instrumentation at the same location will be of interest for readers.

6. **Lines 288-289:** What exactly is meant by "artificial intelligence and neural networks" here? I mean technically NNs are a type of machine learning which is itself a subset of AI (AI gets thrown along as a buzz word quite a bit). I would rephrase this and add a bit more detail.

    1. We have clarified that most of the research involving the SMAS has used neural network based algorithms to do the geometry and riming classification. We have clarified this on lines 291-295. See specific text below as well.

"Multiple studies have shown that neural networks, can be applied to classify snowflakes into six geometrical categories (small particle, planar crystal, columnar crystal, combination of columnar and planar crystal, aggregate, and graupel), as well as five separate degrees of riming can be observed, which is essential for understanding the formation and meteorological environment of snow (Hicks and Notaroš 2019, Key et al. 2021, Thant et al. 2023a)."

7. **Line 313:** This is neat, are there any more details you could provide about the mmWave radar? How sensitive is it and what is its resolution?

   1. Yes, we have included a reference to the manuscript that describes the radar specifications on lines 315-317, as well as added a few more details in the text. See specific text below as well.

"Additionally, it provides remote samples of reflectivity and velocity from an onboard, vertically pointing 74GHz millimeter wave (mmWave) radar with 0.4887m resolution (see Table C.1 of Britto Hupsel de Azevedo 2024 for radar specifications)."

8. **Line 381:** This introductory sentence is a bit awkward, you may want to rework it to flow better.

   1. We have clarified this introductory sentence by rewording it. Please see lines 385-286. See specific text below as well.

"In this section, a list of precipitation events across the three years of deployments are derived from the PARSIVEL[2] to provide a starting point for specific research applications (Appendix A). To accompany the event list, additional in-depth descriptions are given for the quality control or data collection on certain instrumentation."

9. **Lines 445-446:** Do you know what causes this temperature measurement issue?

   1. We believe this is based on a computer issue (see lines 449-450), but we are not positive. We reached out to the manufacturer, Met One, but they also had never seen the issue before.

10. **Figure 15.a:** I find it a bit challenging to interpret this panel with the inner and outer colored scatter. What are the key takeaways from this as we move into a cloudy regime and the PSD widens?

   1. We have added additional information to the description of this figure, specifically due to the necessary inner and outer coloring it uses. This should provide readers with a better understanding of how to interpret the figure. See lines 499-502 or the specific text below as well.

"In the cloud droplet size distribution profile, the outer color represents the distribution's mode at each altitude, that is, the cloud droplet size bin with the highest count while the inner color represents the largest particle size detected at that altitude. Here, the cloud droplet distribution explained what type of cloud region the WxUAS sampled. Near the surface, there were low particle counts and both the inside and outside coloring were blue, so the WxUAS was in clear air (Fig. 15a); as the WxUAS approached 100m, the inner color was red and the outer was blue, which indicates a mix of small and large particles see in cloud transition layer, so the WxUAS likely was starting to enter clouds. Fig. 15b presented a micro-range water content which indicates the vertical variations of water content in the cloud droplet size range."

11. **Section 4:** It wasn't clear to me at first that this was a new Section, I would rework the subsection title here to make that more obvious.

   1. We have added a new title for the section to make it clearer for readers. See line 510 or the specific text below as well.

"4 Highlights and Comparisons of the Instrumentation during the February 28th, 2023, Nor'easter"

12. **Line 572:** Should this be "first half" and not "second half"?

   1. This should be "second half", but the sentence was worded confusingly. We have made corrections to make it clearer on lines 574-576. See specific text below as well.

"The largest snowflakes at the D3R site occurred during the period of heaviest precipitation at the beginning of the event, and the D3R site tended to have slightly larger flakes than the GAIL Site throughout the second half of the event (Fig. 17b). Despite the larger snowflakes at the D3R site later in the event, the snow rates did not differ significantly from the GAIL site (Fig. 17c)."

13. **Section 5:** As a data paper, it would be nice if you could also include a few details/statistics about the dataset itself here (e.g., the data format, layout, size, CF-conventions used for metadata). This doesn't need to be super comprehensive, but might help orient the reader as to how they can interact with the products you've put together.

    1. NASA's GHRC produces user guides associated with each dataset that contains most, if not all, of this information. We have added additional language here clarifying that for readers and what is contained in the document. These are very helpful for users of the dataset, so we are glad to add this information and highlight the work of the GHRC staff. Please see lines 621-627 or below.

"The dataset presented in this article is available through NASA's Global Hydrometeorology Resource Center (GHRC) at the following campaign wide DOI: http://dx.doi.org/10.5067/GPMGVUCONN/DATA101 (Cerrai et al. 2025). Table 4 below includes the individual dataset DOIs. All datasets processed by GHRC goes through standardized quality control checks and documentation as certified by the World Data System CoreTrustSeal (Global Hydrometeorology Resource Center 2024). This data processing done at GHRC includes all steps of a dataset's life cycle from ingestion to processing, archiving, cataloguing, and documenting; their work also supports the distribution on final, published datasets. Accompanying each individual dataset is a user guide that provides additional information and metadata about the dataset including a description of the instrument, file naming conventions, the variables stored within each file, and the data format of the files."

Citations:

Cerrai, D., Wolff, D., Tokay, A., Helms, C., and Filipiak, B.: Global Precipitation Measurement Ground Validation Campaign at the University of Connecticut Collection. Data available online from the NASA EOSDIS Global Hydrometeorology Resource Center Distributed Active Archive Center, Huntsville, Alabama, U.S.A. DOI: http://dx.doi.org/10.5067/GPMGVUCONN/DATA101, 2025.

Global Hydrometeorology Resource Center: 2027-03-14 - Global Hydrometeorology Resource Center - CoreTrustSeal Requirements 2020-2022, DataverseNL, V1, https://doi.org/10.34894/A8ADKU, 2024.

14. **Conclusions Section:** I appreciated that you have some space here aimed towards applications. Sometimes by the end of these data papers you are left wondering, alright, what can we actually use this for? I wonder if including a few additional references for followup applications using similar datasets in previous literature might help provide more concrete ideas for readers to follow from here? Some recent papers that came to mind for this include works like: King et al., 2024 for Lines 635-636, and Billault-Roux et al., 2023 and Shates et al., 2025 for lines 637-638. However, I leave this choice up to the authors.

    1. We have added some additional references, as suggested, in this section to give authors of potential applications for future research ideas. Please see lines 654-659 or the specific text below as well.

"Experiments involving intercomparisons between the ground-based instruments would allow for understanding of how the different types of instrumentation observe the same precipitation. Comparisons between the remote sensing instruments and in-situ instruments are also possible (Billault-Roux et al. 2023; Shates et al. 2025). Other research using this dataset has already been started including improving the PIP's algorithm to accurately identify rain versus snow, refining SWER(ZE) relationships (Tokay et al. 2023; Inglis et al. 2024), used to estimate snow water equivalent rates from radar reflectivity, between the PIP and MRR, understanding microphysical properties of snowfall (Chang et al. 2024; King et al. 2024), and intercomparison of precipitation accumulations from different instrumentation."

15. **Final Comment:** It would also be nice if you could link all the disparate instruments and products together in a single location/repository. It is a pain to have to cycle between different platforms and logins to download things (this also makes it more challenging to find relevant data).

1. NASA GHRC has been working on this, and the DOI and webpage was not available at the time of the manuscript was submitted. This information has been added in Section 5. See lines 619-620 or below.

"The dataset presented in this article is available through NASA's Global Hydrometeorology Resource Center (GHRC) at the following campaign wide DOI http://dx.doi.org/10.5067/GPMGVUCONN/DATA101 (Cerrai et al. 2025). "

Citations:

Cerrai, D., Wolff, D., Tokay, A., Helms, C., and Filipiak, B.: Global Precipitation Measurement Ground Validation Campaign at the University of Connecticut Collection. Data available online from the NASA EOSDIS Global Hydrometeorology Resource Center Distributed Active Archive Center, Huntsville, Alabama, U.S.A. DOI: http://dx.doi.org/10.5067/GPMGVUCONN/DATA101, 2025.

**References**

Billault-Roux, A.-C., Grazioli, J., Delanoë, J., Jorquera, S., Pauwels, N., Viltard, N., Martini, A., Mariage, V., Gac, C. L., Caudoux, C., Aubry, C., Bertrand, F., Schwarzenboeck, A., Jaffeux, L., Coutris, P., Febvre, G., Pichon, J. M., Dezitter, F., Gehring, J., … Berne, A. (2023). ICE GENESIS: Synergetic Aircraft and Ground-Based Remote Sensing and In Situ Measurements of Snowfall Microphysical Properties. Bulletin of the American Meteorological Society, 104(2), E367–E388. https://doi.org/10.1175/BAMS-D-21-0184.1

Casella, D., Panegrossi, G., Sanò, P., Marra, A. C., Dietrich, S., Johnson, B. T., & Kulie, M. S. (2017). Evaluation of the GPM-DPR snowfall detection capability: Comparison with CloudSat-CPR. Atmospheric Research, 197, 64–75. https://doi.org/10.1016/j.atmosres.2017.06.018

King, F., Pettersen, C., Dolan, B., Shates, J., & Posselt, D. (2024). Primary Modes of Northern Hemisphere Snowfall Particle Size Distributions. Journal of the Atmospheric Sciences, 81(12), 2093–2113. https://doi.org/10.1175/JAS-D-24-0076.1

Pierre, A., Jutras, S., Smith, C., Kochendorfer, J., Fortin, V., & Anctil, F. (2019). Evaluation of Catch Efficiency Transfer Functions for Unshielded and Single-Alter-Shielded Solid Precipitation Measurements. Journal of Atmospheric and Oceanic Technology, 36(5), 865–881. https://doi.org/10.1175/JTECH-D-18-0112.1

Shates, J. A., Pettersen, C., L'Ecuyer, T. S., & Kulie, M. S. (2025). KAZR-CloudSat Analysis of Snowing Profiles at the North Slope of Alaska: Implications of the Satellite Radar Blind Zone. Journal of Geophysical Research: Atmospheres, 130(6), e2024JD042700. https://doi.org/10.1029/2024JD042700

Smith, C. D. 2007. "Correcting the Wind Bias in Snowfall Measurements Made with the Geonor T-200B Precipitation Gauge and Alter Wind Shield." Proceedings 4th Symposium on Observations and Instrumentation, American Meteorological Society (AMS) Annual Meeting, San Antonio, Texas.

---

## Author Comment (AC2)

Responses to Referee 2 Comments for ESSD-2025-162: Winter Precipitation Measurements in New England: Results from the Global Precipitation Measurement Ground Validation Campaign in Connecticut

**Referee Comments 2 (RC2):**

General comments:

This submission from B. Filipiak, D. B. Wolff, A. Spaulding, et al. summarizes a field campaign over 3 winter seasons at the University of Connecticut. This project took place from 2021-2024 and deployed several instruments to two sites. This campaign is motivated by the need for validation of remotely-sensed measurements as part of the Global Precipitation Measurement Ground Validation program. The campaign collected data during 117 distinct precipitation events over the 3 winter seasons and this manuscript illustrates how a Nor'easter on February 28, 2023 can be analyzed using this large dataset. Explanations of decision-making for instrument location and caveats with the data quality are made clear.

All DOIs in the Data Availability statement lead to associated links on Earthdata with clear user guides. However, I'm unable to download any datasets which seems likely to be an issue with my login and/or the website and not within the authors' control. I've contacted Earthdata but haven't heard back after several days, so I can't offer a review of the dataset quality and thus rate it "fair."

Overall, the manuscript and field campaign are scientifically interesting and novel. I recommend this manuscript for acceptance to ESS-D with minor revisions, so long as the editor and other reviewers are able to access the datasets.

We thank the reviewer for their feedback, and we have addressed all their comments below. We apologize for the confusion on accessing the data through the EarthData portal. We have confirmed with our contact at NASA's GHRC that all datasets are publicly available. Please note that the lines mentioned in the responses refer to the clean copy of the revised manuscript.

Specific comments:

1. The text in multiple figures (Figures 14 onward) could be enlarged.

    1. We have made changes to these figures, including increasing the font size, in order to improve their clarity. Please see the new figures on lines 456 (Fig 14), 506 (Fig 15), 563 (Fig 16), 609 (Fig 17) or below.

[Figure]

Figure 14

[Figure]

Figure 15

[Figure]

Figure 16

[Figure]

Figure 17

2. L222: "first two size bin" is this wording correct? Maybe I'm misunderstanding what's being stated here.

    1. While this wording is correct, we see that it was slightly confusing. We have updated the wording to improve the clarity. Please see lines 205-206 or the see specific text below as well.

"Due to sensitivity of laser device to the small particles with minimum detectable size 0.25 mm in equivalent diameter, the first two size bins in the PARSIVEL2 observations are always empty."

3. Figure 7: misspelling in caption, "ACHIVE"

    1. We have made this change to the figure caption on line 263. See the text below as well.

"Fig. 7. Deployment of the ACHIEVE trailer with the W-Band Radar on the roof during 2022-2023 at the D3R site."

4. Table 1: In caption, "the superscript D in the 2022-2023 column indicates the instrument was only at the GAIL site" should be the D3R site.

    1. We have made this change to the table caption on lines 376-379. See the text below as well.

"Table 1. List of instruments deployed during the three winters of the NASA GPM GV field campaign at UConn. The * in the 2022-2023 column indicates the instrument was deployed both at the GAIL and D3R sites; the superscript G in the 2022-2023 column indicates the instrument was only at the GAIL site; the superscript D in the 2022-2023 column indicates the instrument was only at the D3R site. The (x2) in the 2023-2024 column indicates there were two instruments deployed at the GAIL site."

5. L583: For consistency, use "Between 0-9 UTC" instead of "Z"

    1. We have made this change on line 593. See the text below as well.

"Between 0-9 UTC, the reflectivity peaks from the W-band (Fig. 17j) generally matched the MRR, even if they had differing intensities. Because the W-band transmitter was more powerful, it better highlighted higher cloud structures during the beginning of event, although the W-band signal attenuates more as precipitation rates increase."

6. L 589: Delete ")" at the end of this line

    1. We have made this change on line 599. See the text below as well.

"During the period of heaviest precipitation, pockets of larger LDR values (light blue shading) are most likely associated with larger aggregates and rimed snow particles."